# LLM Unlearning via Loss Adjustment with only Forget Data

**Yaxuan Wang**[*1]  **Jiaheng Wei**[†3]  **Chris Yuhao Liu**[1]  **Jinglong Pang**[1]  **Quan Liu**[2]
**Ankit Parag Shah**[2]  **Yujia Bao**[2]  **Yang Liu**[1]  **Wei Wei**[2]
[1]University of California Santa Cruz   [2]Center for Advanced AI, Accenture
[3]The Hong Kong University of Science and Technology (Guangzhou)

## Abstract

Unlearning in Large Language Models (LLMs) is essential for ensuring ethical and responsible AI use, especially in addressing privacy leak, bias, safety, and evolving regulations. Existing approaches to LLM unlearning often rely on retain data or a reference LLM, yet they struggle to adequately balance unlearning performance with overall model utility. This challenge arises because leveraging explicit retain data or implicit knowledge of retain data from a reference LLM to fine-tune the model tends to blur the boundaries between the forgotten and retain data, as different queries often elicit similar responses. In this work, we propose eliminating the need to retain data or the reference LLM for response calibration in LLM unlearning. Recognizing that directly applying gradient ascent on the forget data often leads to optimization instability and poor performance, our method guides the LLM on what not to respond to, and importantly, how to respond, based on the forget data. Hence, we introduce **F**orget data only **L**oss **A**djustmen**T** (**FLAT**), a "flat" loss adjustment approach which addresses these issues by maximizing $f$-divergence between the available template answer and the forget answer only w.r.t. the forget data. The variational form of the defined $f$-divergence theoretically provides a way of loss adjustment by assigning different importance weights for the learning w.r.t. template responses and the forgetting of responses subject to unlearning. Empirical results demonstrate that our approach not only achieves superior unlearning performance compared to existing methods but also minimizes the impact on the model's retained capabilities, ensuring high utility across diverse tasks , including copyrighted content unlearning on Harry Potter dataset and MUSE Benchmark, and entity unlearning on the TOFU dataset [1].

## 1 Introduction

The widespread integration of Large Language Models (LLMs) into daily applications has raised significant concerns regarding the trustworthiness of such models. Their outputs may contain sensitive, private, or illegal content (Karamolegkou et al., 2023; Patil et al., 2023), reflect societal biases (Motoki et al., 2024; Yu et al., 2023), or provide harmful instructions (Yao et al., 2023; Li et al., 2024; Barrett et al., 2023). In particular, for privacy concerns, regulations (Hoofnagle et al., 2019) have been introduced, requiring applications to support the deletion of information contained in training samples upon user request. This has motivated research into machine unlearning (MU) (Cao & Yang, 2015; Liu et al., 2024b; Fan et al., 2023; Di et al., 2024; Liu et al., 2024c), a critical process aimed at removing the influence of specific data points, data classes, or even higher-level data concepts from trained models.

LLM unlearning (Eldan & Russinovich, 2023; Yao et al., 2023; Liu et al., 2024c) is part of a broader set of MU techniques aiming to make the unlearned model forget the knowledge specified in forget dataset, while preserving the model ability to accomplish tasks irrelevant to the unlearning target (Liu

---

[*]Work done during Yaxuan Wang's internship at Center for Advanced AI, Accenture.

[†]Work mainly done at Center for Advanced AI, Accenture, corresponding to jiahengwei@hkust-gz.edu.cn.

[1]The code is available at `https://github.com/UCSC-REAL/FLAT`.

Table 1: **Comparison of different loss adjustment-based baselines in terms of their requirement.** Our method relies solely on forget data and available template responses, without using the retain data or a reference model for response calibration.

| Baselines | Forget Data | Retain Data | Reference Model |
|---|---|---|---|
| Gradient Ascent (GA) (Maini et al., 2024a) | ✓ | ✗ | ✗ |
| Gradient Difference (GD) (Maini et al., 2024a) | ✓ | ✓ | ✗ |
| KL Minimization (KL) (Maini et al., 2024a) | ✓ | ✓ | ✓ |
| Preference Optimization (PO) (Maini et al., 2024a) | ✓ | ✓ | ✗ |
| Mismatch (Liu et al., 2024a) | ✓ | ✓ | ✗ |
| Direct Preference Optimization (DPO) (Rafailov et al., 2024) | ✓ | ✗ | ✓ |
| Negative Preference Optimization (NPO) (Zhang et al., 2024) | ✓ | ✗ | ✓ |
| Large Language Model Unlearning (LLMU) (Yao et al., 2023) | ✓ | ✓ | ✓ |
| **FLAT** (Ours) | ✓ | ✗ | ✗ |

et al., 2024c; Ji et al., 2024). To achieve this, existing work can be categorized into three main streams of LLM unlearning approaches: input-based, data-based, and model-based methods. Input-based methods (Liu et al., 2024a; Pawelczyk et al., 2023) design input instructions to guide the original LLM towards the unlearning objective without altering the model's parameters. Data-based methods (Choi et al., 2024) typically fine-tune models on pre-constructed desirable responses, using prompts from the forget data distribution. Model-based methods (Yao et al., 2023; Chen & Yang, 2023) focus on modifying the weights or architecture to achieve the unlearning objective. Among these approaches, the most relevant to our work is fine-tuning the target LLM using a modified loss function, which typically incorporates two key objectives: maximizing the loss on the forget samples and minimizing (or maintaining) the loss on the retain samples.

However, as summarized in Table 1, current loss adjustment-based methods either rely on retain data (Maini et al., 2024a; Liu et al., 2022; Yao et al., 2023), which might not be readily available in real-world scenarios (Li et al., 2024), or utilize a reference model (Rafailov et al., 2024; Zhang et al., 2024; Yao et al., 2023; Maini et al., 2024a) to maintain performance on the retain dataset, incurring additional cost during training—especially when fine-tuning a large-scale LLM. Moreover, leveraging explicit retain data or implicit knowledge from a reference LLM during fine-tuning may blur the distinction between the forget and retain data, which can lead to a trade-off between model utility and forget quality. Furthermore, fine-tuning using both retain data and forget data would require a careful design of a data mixing strategy. To preserve model utility while improving forget quality, we propose **F**orget data only **L**oss **A**djustmen**T** (**FLAT**), a "flat" loss adjustment approach which adjusts the loss function using only the forget data. Given the forget data, **FLAT** guides the LLM not only in what to forget but also in how to respond, by optimizing the $f$-divergence between the template and forget answers with respect to the forget data. The variational form of the $f$-divergence enables loss adjustment by assigning optimal importance weights to learning from template responses while forgetting the responses subject to unlearning. Our main contributions are highlighted below:

- We identify the potential drawback of relying on retain data or a reference LLM to guide LLM unlearning. To address this, we propose **FLAT**, which facilitates LLM unlearning without requiring retain data or a reference LLM for response calibration.

- **FLAT** optimizes the $f$-divergence between template and forget responses to guide the LLM through the unlearning process. The variational form of $f$-divergence optimization provides a clear illustration of how to optimally balance forget quality and model utility, with theoretical guarantees.

- Extensive experiments on three unlearning tasks, including copyrighted content unlearning on the Harry Potter dataset and MUSE benchmark, as well as entity unlearning on the TOFU dataset, demonstrate the superior performance of our method, achieving both high unlearning efficiency and strong overall model utility.

## 2 PRELIMINARIES

In this section, we introduce the preliminary formulation of LLM unlearning and existing LLM unlearning framework.

## 2.1 FORMULATION

Given an forget dataset $D_f$, a retain dataset $D_r$, and an LLM $\theta_o$, the task of LLM unlearning is to fine-tune the original model such that the updated LLM $\theta$ resembles a model trained without $D_f$. For a prompt-response pair $(x, y)$, the loss function on $y$ for fine-tuning is $\mathcal{L}(x, y; \theta) = \sum_{i=1}^{|y|} \ell(h_\theta(x, y_{<i}), y_i)$, where $\ell(\cdot)$ is the cross-entropy loss, and $h_\theta(x, y_{<i}) := \mathbb{P}(y_i|(x, y_{<i}); \theta)$ is the predicted probability of the token $y_i$ given by an LLM $\theta$, with the input prompt $x$ and the already generated tokens $y_{<i} := [y_1, ..., y_{i-1}]$.

The most straightforward approach to unlearn is Gradient Ascent (GA). GA modifies a trained model so that it "forgets" or minimizes the influence of specific data or patterns it has previously learned. Mathematically, the GA algorithm iteratively updates the model at step $t$ by performing gradient ascent on the next-token prediction loss over the forget dataset: $\theta_{t+1} \leftarrow \theta_t + \lambda \nabla_{\theta_t} \mathcal{L}(x, y; \theta_t)$, where $\lambda$ is the (un)learning rate.

## 2.2 EXISTING LLM UNLEARNING PARADIGM

The mainstream class of existing LLM unlearning methods involves fine-tuning the original LLM against an unlearning objective function. Although the exact designs vary, the general type of loss adjustment in LLM unlearning can be characterized as follows:

$$L = L_{\mathbf{FG}} + L_{\mathbf{RT}} + L_{\mathbf{Custom}}. \tag{1}$$

The modified loss function comprises three main components:

- $L_{\mathbf{FG}}$ (**Forget Loss**): Encourages the model to "forget" the undesired data or patterns. This typically involves increasing the loss on the data to be forgotten, effectively making the model perform worse on those specific examples. The goal is to reduce the model's reliance on these data points, thereby minimizing their influence on future predictions.

- $L_{\mathbf{RT}}$ (**Retain Loss**): Ensures that the model maintains its overall performance and general knowledge on unaffected data. It typically involves using the original loss function from training or a modified version that focuses on the data the model is meant to retain. This term prevents the unlearning process from degrading the model's overall capabilities beyond the scope of the specific unlearning objective.

- $L_{\mathbf{Custom}}$ (**Custom Loss**): Allows for additional flexibility and customization in the unlearning process. It may include regularization terms to control the magnitude of parameter updates or specific constraints to enforce certain unlearning behaviors. This component enables researchers to tailor the unlearning process to specific requirements or incorporate domain-specific knowledge.

In summary, common loss adjustment methods employ one (Jang et al., 2022), two (Liu et al., 2022; Maini et al., 2024a; Zhang et al., 2024), or all three (Yao et al., 2023) of these components to guide the model towards forgetting specific data while minimizing the impact on its overall performance and utility. The interplay between these terms allows for controlled and targeted unlearning, ensuring the model retains its valuable capabilities while selectively forgetting undesired information. More detailed formulations of these loss adjustment-based methods, along with related work, are deferred to Appendix C.1 and Appendix E.

**An Example: Large Language Model Unlearning (LLMU).** We adopt a popular approach in LLM unlearning, LLMU (Yao et al., 2023), to interpret a special case of Eqn. (1). Specifically, the objective of LLMU contains three components: the Unlearn Harm $L_{\mathbf{FG}}$, the Maintain Performance $L_{\mathbf{RT}}$, and the Random Mismatch $L_{\mathbf{Random}}$ (the custom loss). The training objective is as follows:

$$L_{\mathbf{LLMU}} = L_{\mathbf{FG}} + L_{\mathbf{RT}} + L_{\mathbf{Random}},$$

The forget loss $L_{\mathbf{FG}} = -\sum_{(x_f, y_f) \in D_f} \mathcal{L}(x_f, y_f; \theta)$, where $(x_f, y_f)$ indicates the forget data pairs from the forget dataset $D_f$, $\theta$ is the updated unlearned model. It is actually the Gradient Ascent loss to forget the samples subject to unlearning. The retain loss $L_{\mathbf{RT}} = \sum_{(x_r, y_r) \in D_r} \sum_{i=1}^{|y_r|} KL(h_{\theta_o}(x_r, y_{r<i}) || h_\theta(x_r, y_{r<i}))$, where $KL(\cdot)$ is the KL divergence term, $(x_r, y_r)$ indicates the retain data pairs from the retain dataset $D_r$, $\theta_o$ is the original model, and $\theta$ is the updated model. The random loss $L_{\mathbf{Random}} = \sum_{(x_f, \cdot) \in D_f} \frac{1}{|Y_{rdn}|} \sum_{y_{rdn} \in Y_{rdn}} \mathcal{L}(x_f, y_{rdn}; \theta)$. Here, $Y_{rdn}$ is a set of random responses that do not have a connection to the forget prompts $x_f$.

## 3    METHOD

In this section, we introduce **F**orget data only **L**oss **A**justmen**T** (**FLAT**), a "flat" loss adjustment approach which adjusts the loss function using only the forget data, by leveraging $f$-divergence maximization towards the distance between the preferred template and original forget responses. We first derive the formulation of our method via $f$-divergence maximization (§ 3.1), followed by the presentation of the empirical alternative to our approach (§ 3.2). In section § 3.3, we explore the estimation gap between the theoretical and empirical $f$-divergence. Finally, we discuss the connection between our method and DPO (§ 3.4).

### 3.1    LOSS-ADJUSTMENTS VIA $f$-DIVERGENCE MAXIMIZATION

For each learning batch, we assume that we only have access to a set of forget samples $(x_f, y_f) \in D_f$. Instead of directly adopting gradient ascent over these forget samples, we propose to maximize the divergence between exemplary and bad generations of forget data. Key steps are summarized as below.

Table 2: $f_{div}$s, optimal variational $g$ ($g^*$), conjugate functions ($f^*$).

| Name | $g^*(v)$ | $\mathrm{dom}_{f^*}$ | $f^*(u)$ |
|---|---|---|---|
| Total Variation | $\dfrac{1}{2} \tanh v$ | $u \in [-\dfrac{1}{2}, \dfrac{1}{2}]$ | $u$ |
| Jensen-Shannon | $\log \dfrac{2}{1 + e^{-v}}$ | $u < \log 2$ | $-\log(2 - e^u)$ |
| Pearson | $v$ | $\mathbb{R}$ | $\dfrac{1}{4}u^2 + u$ |
| KL | $v$ | $\mathbb{R}$ | $e^{u-1}$ |

- **Step 1:** Equip example/template responses $y_e$ for each forget sample $x_f$. Together we denote the paired samples as $\{(x_f^j, y_e^j)\}_{j \in [N]}$.

  This could be done by leveraging open-source LLMs such as Llama 3.1 (Dubey et al., 2024) or self-defining the responses according to our wish, etc. The designated unlearning response could be a reject-based answer such as "I don't know" (denoted as "IDK") or an irrelevant answer devoid of the unlearning target-related information.

  **Motivation:** Step 1 generates example responses for LLM fine-tuning and provides better instructions on what LLM should respond given the forget data. Besides, certain existing methods make LLM generate hallucinated responses after unlearning, which further illustrates the importance of example responses for LLM unlearning.

- **Step 2:** Loss adjustmens w.r.t. the sample pairs $(x_f, y_e, y_f)$ through:

$$L(x_f, y_e, y_f; \theta) = \lambda_e \cdot L_e(x_f, y_e; \theta) - \lambda_f \cdot L_f(x_f, y_f; \theta), \qquad (2)$$

  where $L_e, L_f$ are losses designed for the data sample $(x_f, y_e)$ and $(x_f, y_f)$, respectively. The corresponding closed form will be introduced in Section § 3.2.
  **Motivation:** Step 2 encourages the LLM to forget the forget data with bad responses, meanwhile, learn to generate good responses on relevant forget data. [such as template answers]

- **Step 3:** How to decide on the values of $\lambda_e$ and $\lambda_f$?
  We leverage $f$-divergence to illustrate the appropriate balancing between $L_e(x_f, y_e; \theta)$ and $L_f(x_f, y_f; \theta)$. Assume $x_f, y_e$ is generated by the random variable $X_f, Y_e$ jointly following the distribution $\mathcal{D}_e$. Similarly, $x_f, y_f$ is given by $X_f, Y_f$ and $(X_f, Y_f) \sim \mathcal{D}_f$. Step 2 shares similar insights as if we are maximizing the divergence between $\mathcal{D}_e$ and $\mathcal{D}_f$. Our theoretical purpose is to obtain the model that maximizes the $f$-divergence between $\mathcal{D}_e$ and $\mathcal{D}_f$, defined as $f_{div}(\mathcal{D}_e || \mathcal{D}_f)$.

  **The variational form $f$-divergence**   Instead of optimizing the $f_{div}$ term directly, we resolve to the variational form of it. Due to the Fenchel duality, we would have:

$$f_{div}(\mathcal{D}_e || \mathcal{D}_f) = \sup_g \left[ \mathbb{E}_{\mathcal{Z}_e \sim \mathcal{D}_e} [g(\mathcal{Z}_e)] - \mathbb{E}_{\mathcal{Z}_f \sim \mathcal{D}_f} [f^*(g(\mathcal{Z}_f))] \right] := \sup_g \mathrm{VA}(\theta, g), \qquad (3)$$

  we define $f^*$ as the conjugate function of the $f$-divergence function. Here, $\mathcal{Z}_e$ takes $(x_f, y_e, \theta)$ as input and estimates the "loss" between the model's response to $x_f$ and the target $y_e$. Mathematically, this corresponds to the discrepancy between $\theta(x_f)$ and $y_e$, where $\theta(x_f)$ represents the answer generated by the LLM parameterized by $\theta$ given prompt $x_f$. Similarly, $\mathcal{Z}_f$ estimates the "loss"

for $(x_f, y_f, \theta)$. We will provide the empirical estimation for Eqn. (3) in the next section. For simplicity, we define $\text{VA}(\theta, g^*) := \sup_g \text{VA}(\theta, g)$, where $g^*$ is the optimal variational function. Hence, the objective of **FLAT** is to obtain: $\theta^* := \arg\max_\theta \text{VA}(\theta, g^*)$.

**Motivation:** Note that existing solutions fail to keep a good balance between model performance on forget data and retain data, step 3 provides a formal theoretical framework of our loss revision in Step 2, under the $f-$divergence maximization between $\mathcal{D}_e$ and $\mathcal{D}_f$, the method assigns the appropriate weights $f^*(\cdot), g^*(f^*(\cdot))$ w.r.t. the joint data distributions $\mathcal{D}_e, \mathcal{D}_f$.

## 3.2 EMPIRICAL ALTERNATIVE OF LOSS ADJUSTMENT

Note that Eqn. (3) could be viewed as a data distribution level loss adjustment, in practice, when given access to a set of forget data as well as example and bad answers, $x_f, y_e, y_f$, the per-sample loss function (closed form of Eqn. (2)) would be given by[2]:

$$
\begin{aligned}
L(x_f, y_e, y_f; \theta) &= - \left[ \sup_g \left[ g(\mathbb{P}(x_f, y_e; \theta)) - f^*(g(\mathbb{P}(x_f, y_f; \theta))) \right] \right] \\
&= -g^*(\mathbb{P}(x_f, y_e; \theta)) + f^*(g^*(\mathbb{P}(x_f, y_f; \theta))).
\end{aligned}
\tag{4}
$$

We provide examples of $f$-divergence functions in Table 2, along with their conjugate and variational functions (Nowozin et al., 2016; Wei & Liu, 2021). We illustrate via following examples.

**Example 1: Total-Variation** For Total-Variation (TV), an example of $f$-divergence, $f^*(u) = u, g^*(v) = \frac{\tanh(v)}{2}$, hence, $g^*(\mathbb{P}(x_f, y_e; \theta)) - f^*(g^*(\mathbb{P}(x_f, y_f; \theta))) = \frac{\tanh(\mathbb{P}(x_f, y_e; \theta))}{2} - \frac{\tanh(\mathbb{P}(x_f, y_f; \theta))}{2}$. We defer examples of other $f$-divergence functions in the Appendix B.1.

**How to estimate** $\mathbb{P}(x_f, y_e; \theta), \mathbb{P}(x_f, y_f; \theta)$**?** We define the following two quantities:

$$
\mathbb{P}(x_f, y_e; \theta) := \frac{\sum_{i=1}^{|y_e|} P(\mathcal{M}_\theta(x_f, y_{e,<i}) = y_{e,i})}{|y_e|}, \quad \mathbb{P}(x_f, y_f; \theta) := \frac{\sum_{i=1}^{|y_f|} P(\mathcal{M}_\theta(x_f, y_{f,<i}) = y_{f,i})}{|y_f|}.
$$

Here, $y_{e,i}$ and $y_{f,i}$ denote the $i$-th token in the samples $y_e$ and $y_f$, respectively, while $y_{e,<i}$ and $y_{f,<i}$ represent the already generated tokens. $|y_e|$ and $|y_f|$ are the lengths of the example response $y_e$ and the forget response $y_f$, respectively. Given a prompt and the previously generated tokens, $P(\mathcal{M}_\theta(x_f, y_{e,<i}) = y_{e,i})$ and $P(\mathcal{M}_\theta(x_f, y_{f,<i}) = y_{f,i})$ are the probabilities of correctly predicting the next token, where $\mathcal{M}_\theta(x_f, y_{e,<i})$, and $\mathcal{M}_\theta(x_f, y_{f,<i})$ are the predicted token using LLM $\theta$ given input prompt $x_f$ and the already generated tokens $y_{e,<i}$ and $y_{f,<i}$. These two quantities represent the average probabilities of correctly generating tokens for the template and forget responses, respectively. To align Eqn. (4) with Eqn. (2), we could define $\lambda_e = \lambda_f = 1$, $L_e(x_f, y_e; \theta) := -g^*(\mathbb{P}(x_f, y_e; \theta))$, and $L_f(x_f, y_f; \theta) := -f^*(g^*(\mathbb{P}(x_f, y_f; \theta)))$.

## 3.3 THE UPPER BOUND OF THE ESTIMATION GAP

To connect the empirical alternative of **FLAT** with the corresponding theoretical format, in this section, we aim to explore the estimation gap between the theoretical $f$-divergence $f_{div}(\mathcal{D}_e||\mathcal{D}_f)$ and the empirical optimal estimated $f$-divergence $\hat{f}_{div}(D_e||D_f)$, here we define:

$$
\hat{f}_{div}(D_e||D_f) := \mathbb{E}_{Z_e \sim D_e}[\hat{g}(Z_e)] - \mathbb{E}_{Z_f \sim D_f}[f^*(\hat{g}(Z_f))],
$$

where $\hat{g} := \sup_{g \in \Phi} \mathbb{E}_{Z_e \sim D_e}[g(Z_e)] - \mathbb{E}_{Z_f \sim D_f}[f^*(g(Z_f))]$ and $\Phi$ is the function space.

**Assumption 3.1** (Bounded Density Ratio). *The density ratio $Z_e/Z_f$ is lower and upper bounded by positive constants $a$ and $b$, respectively.*

Assumption 3.1 is wildely adopted by the literature (Suzuki et al., 2008; Nguyen et al., 2010), which necessitates that the probability density functions $Z_e, Z_f$ share the same support.

---

[2]To clarify, we introduce the negative sign on the r.h.s. because loss function is commonly combined with the minimization task, while our method is formulated as maximizing the $f$-divergence.

**Assumption 3.2** (Regularity of Divergence Function). *$f(\cdot)$ is smooth on $[a, b]$, and $f(1) = 0$. $f$ is $\mu_0$-strongly convex, and has $L_0$-Lipschitz continuous gradient on $[a, b]$, for positive constants $\mu_0, L_0$.*

Assumption 3.2 is a mild condition since it only requires the condition to hold for the interval $[a, b]$, which works for many commonly used $f$-divergence functions, i.e., KL divergence.

Let $(V, ||\cdot||_{L_2})$ be a normed space, and $\Phi \subset V$. $v_1, ..., v_C$ is a $\delta$-covering over $\Phi$ of size $C$ if $\Phi \subset \cup_{i=1}^{C} B(v_i, \delta)$ where $B(v_i, \delta)$ is the $\delta$-ball centered at $v_i$. The covering number is then defined as $C_2(\delta, \Phi) = \min\{C : \exists \delta\text{-covering over } \Phi \text{ of size } C\}$. The following assumption would characterize the representation power of the function space $\Phi$.

**Assumption 3.3** (Order of Covering Number). *$C_2(\delta, \Phi) = \mathcal{O}(\exp\{\delta^{-r_\Phi}\})$, and $r_\Phi \in (0, 2)$.*

**Theorem 3.4.** *Given Assumptions 3.1-3.3, suppose $\hat{g} \in \Phi$, with probability $\geq 1 - e^{-N^{r_\Phi/(2+r_\Phi)}}$, we have: $\left| \hat{f}_{div}(D_e || D_f) - f_{div}(\mathcal{D}_e || \mathcal{D}_f) \right| \precsim N^{-\frac{1}{r_\Phi+2}}$, where we have defined $N$ as the number of samples in the forget data.*

Theorem 3.4 illustrates that the empirical alternative of **FLAT**, $\hat{f}_{div}(D_e || D_f)$, achieves the optimal non-parametric rate of convergence towards $f_{div}(\mathcal{D}_e || \mathcal{D}_f)$.

## 3.4 CONNECTION WITH DPO

In this section, we discuss the connection and key differences between our approach and the celebrated Direct Preference Optimization (DPO) Rafailov et al. (2024) approach for aligning LLMs.

Given a dataset $D = \{(x_f^j, y_e^j, y_f^j)\}_{j \in [N]}$, where $y_e$ and $y_f$ are preferred template and original forget responses to the forget prompt $x_f$, DPO (Rafailov et al., 2024) fine-tunes original model $\theta_o$ using $D$ to better align it with good answer preferences, which minimizes:

$$L_{\text{DPO},\beta}(\theta) = -\frac{2}{\beta} \mathbb{E}_D \left[ \log \sigma \left( \beta \log \frac{\pi_\theta(y_e \mid x_f)}{\pi_{ref}(y_e \mid x_f)} - \beta \log \frac{\pi_\theta(y_f \mid x_f)}{\pi_{ref}(y_f \mid x_f)} \right) \right]$$

$$= -\frac{2}{\beta} \mathbb{E}_D \left[ \log \sigma \left( \beta (\log \prod_{i=1}^{|y_e|} h_\theta(x_f, y_{e,<i}) - \log \prod_{i=1}^{|y_f|} h_\theta(x_f, y_{f,<i})) - M_{ref} \right) \right],$$

where, $\sigma(t) = \frac{1}{1+e^{-t}}$ is the sigmoid function, $\beta > 0$ is the inverse temperature, $\pi_\theta := \prod_{i=1}^{|y|} h_\theta(x, y_{<i})$ is the predicted probability of the response $y$ to prompt $x$ given by LLM $\theta$, $\pi_{ref}$ is the predicted probability given by reference model, and $M_{ref} := \beta (\log \prod_{i=1}^{|y_e|} h_{\theta_o}(x_f, y_{e,i}) - \log \prod_{i=1}^{|y_f|} h_{\theta_o}(x_f, y_{f,i}))$. As for **FLAT**, we calculate the average probability of all correctly generated tokens and employ a novel re-weighting mechanism that assigns different importance to each term using distinct activate functions for both the example and forget loss terms, which minimizes:

$$L_{\text{FLAT}}(\theta) = -\mathbb{E}_D \left[ g^*(\frac{1}{|y_e|} \sum_{i=1}^{|y_e|} h_\theta(x_f, y_{e,<i})) - f^*(g^*(\frac{1}{|y_f|} \sum_{i=1}^{|y_f|} h_\theta(x_f, y_{f,<i}))) \right].$$

Here, $f^*(\cdot), g^*(f^*(\cdot))$ are the activate functions that assign appropriate weights to each loss term. The detailed derivation is in Appendix B.2. The key differences are highlighted in red. Specifically, DPO relies on a reference model to guide the unlearning process, whereas **FLAT** only uses a sample pair dataset containing both exemplar and forget responses. Besides, our solution differs from DPO in three critical aspects: the re-weighting activation function, whether to sum or average the token losses, and whether to apply the logarithm to the output probability. We conduct an ablation study with DPO to evaluate the effectiveness of the proposed re-weighting mechanism in Section § 4.5.

## 4 EXPERIMENT

In this section, we compare the proposed method with baseline unlearning methods on three widely used LLM unlearning tasks: copyrighted content unlearning on Harry Potter (HP) Series Book (Yao et al., 2023) (§ 4.2), entity unlearning on TOFU dataset (Maini et al., 2024a) (§ 4.3), and unlearning on MUSE-News benchmark (Shi et al., 2024) (§ 4.4). We conduct additional ablation studies to assess the effectiveness of our methods in Section § 4.5.

## 4.1 BASELINE METHODS

We evaluate the effectiveness of our proposed method **FLAT** by comparing it to a series of strong LLM unlearning baselines, particularly those based on loss adjustment. We consider Gradient Ascent (GA) (Jang et al., 2022; Yao et al., 2023), KL minimization (KL) (Maini et al., 2024a), GradDiff (GD) (Liu et al., 2022), NPO (Zhang et al., 2024), and Mismatch (Liu et al., 2024a) across all three tasks. For copyrighted content and entity unlearning, we also include Preference Optimization (PO) (Maini et al., 2024a), Large Language Model Unlearning (LLMU) (Yao et al., 2023), and DPO (Rafailov et al., 2024). For the MUSE-News benchmark, we additionally consider Task Vectors (Ilharco et al., 2022), Who's Harry Potter (WHP) (Eldan & Russinovich, 2023) and an extended version of NPO (NPO-RT) as a comparable method, which incorporates a fine-tuning term on the retain dataset. Further experiment details are provided in Appendix C.1.

## 4.2 COPYRIGHTED CONTENT UNLEARNING

**Experiment Setup.** We select Harry Potter and the Sorcerer's Stone (Rowling, 1997) as the copyrighted content for unlearning. The objective is to ensure that the unlearned model does not generate passages with high similarity to the original text. Following prior works (Liu et al., 2024a; Yao et al., 2023), we first fine-tune LLMs on the corresponding corpus, treating it as the model subject to unlearning, while using the original pre-trained checkpoint as the retained model[3]. Following Yao et al. (2023); Jia et al. (2024), We extract 400 chunks from the Harry Potter book series dataset (Eldan & Russinovich, 2023), with each chunk containing up to 512 tokens, to create the forget dataset $D_f$. We sample 400 paragraphs in the C4 dataset (Raffel et al., 2020) as the retain data $D_r$. The IDK dataset comes from Jia et al. (2024). We experiment with OPT-2.7B (Zhang et al., 2022) and Llama2-7B (Touvron et al., 2023) for this task.

**Evaluation Metrics.** We report three key metrics to assess the unlearning efficiency and model utility of the unlearned models. For unlearning efficiency, we use the Forget Quality Gap (FQ Gap), similar to Liu et al. (2024a), which is the sum of the BLEU Gap

Table 3: Performance of our method and the baseline methods on Harry Potter dataset using OPT-2.7B. **FLAT** consistently ranks in the top two in terms of similarity to the retained model, measured by Forget Quality Gap (FQ Gap), while also generating meaningful and diverse outputs, as reflected by perplexity (PPL) and the average zero-shot accuracy across nine LLM benchmarks (Avg. Acc.). The top two results across three main metrics are highlighted in **blue**.

| Metric | FQ Gap($\downarrow$) | PPL($\downarrow$) | Avg.Acc.($\uparrow$) |
|---|---|---|---|
| Original LLM | 1.5346 | 15.6314 | 0.4762 |
| Retained LLM | 0.0 | 14.3190 | 0.4686 |
| GA | 2.7301 | 1.0984e71 | 0.3667 |
| KL | 2.7301 | 16.1592 | 0.4688 |
| GD | 2.3439 | 16.1972 | 0.4690 |
| PO | 2.1601 | **14.8960** | 0.4583 |
| Mismatch | 1.4042 | 15.7507 | 0.4679 |
| LLMU | 2.4639 | 15.8398 | 0.4656 |
| DPO | 2.2152 | 16.8396 | 0.4621 |
| NPO | **1.2611** | 19.6637 | 0.4644 |
| **FLAT** (TV) | 1.4047 | 15.5512 | 0.4681 |
| **FLAT** (KL) | **1.3238** | **15.5311** | **0.4694** |
| **FLAT** (JS) | 1.4025 | 15.5499 | **0.4693** |
| **FLAT** (Pearson) | 1.4089 | 15.5543 | 0.4686 |

and ROUGE-L Gap. It is the absolute difference between the retained model and the unlearned model on these metrics. Specifically, we calculate BLEU (Papineni et al., 2002) and ROUGE-L (Lin, 2004) scores by comparing ground-truth excerpts with completions generated by the unlearned model, given a fixed prefix length of 200 tokens from the forget data, to reflect potential copyright content leakage. We further conduct study on the prompt length for evaluation in Appendix D.1. Following (Ji et al., 2024), we measure the model utility using the zero-shot accuracy on nine standard LLM benchmarks to determine if the generated text remains meaningful and diverse. Additionally, we measure perplexity (PPL) on Wikitext (Merity et al., 2016). More details about the experimental setup and implementation are in Appendix C.2.

**FLAT consistently ranks in the top two across three metrics.** Table 3 shows that **FLAT** ranks in the top two across the three primary metrics, with particularly strong performance in KL f-divergence function. Our method achieves scores close to those of the retained model in terms of the average accuracy across nine LLM benchmarks.

**FLAT approach achieves good trade-off.** Our method demonstrates strong unlearning efficiency while preserving model utility as shown in Table 3. Although NPO demonstrates the best forget quality, outperforming our method, it severely suffers from lower model utility, as reflected by its PPL score. PO, while also using example responses, has the lowest PPL and a weak forgetting

---

[3]We empirically verified that the initial LLM cannot generate the original corpus, making it a valid candidate for retrained model.

Table 4: Performance of our method and the baseline methods on TOFU dataset using three base LLMs, Llama2-7B, Phi-1.5B, and OPT-2.7B. FQ, MU, R-RL, F-RL represent forget quality, model utility, ROUGE-L on retain dataset and ROUGE-L on forget dataset respectively. We include the original LLM and retain LLM for reference. The top two results are highlighted in **blue**.

| Base LLM | Llama2-7B | | | | Phi-1.5B | | | | OPT-2.7B | | | |
|---|---|---|---|---|---|---|---|---|---|---|---|---|
| Metric | FQ | MU | F-RL(↓) | R-RL | FQ | MU | F-RL(↓) | R-RL | FQ | MU | F-RL(↓) | R-RL |
| Original LLM | 4.4883e-06 | 0.6346 | 0.9851 | 0.9833 | 0.0013 | 0.5184 | 0.9607 | 0.9199 | 0.0013 | 0.5120 | 0.7537 | 0.7494 |
| Retained LLM | 1.0 | 0.6267 | 0.4080 | 0.9833 | 1.0 | 0.5233 | 0.4272 | 0.9269 | 1.0 | 0.5067 | 0.4217 | 0.7669 |
| GA | 0.0143 | 0.6333 | 0.4862 | 0.9008 | 0.0013 | 0.5069 | 0.5114 | 0.8048 | 0.2657 | 0.4639 | 0.4748 | 0.6387 |
| KL | 0.0068 | 0.6300 | 0.5281 | 0.9398 | 0.0030 | 0.5047 | 0.5059 | 0.8109 | 0.0286 | 0.4775 | 0.4810 | 0.6613 |
| GD | 0.0068 | 0.6320 | 0.4773 | 0.8912 | 0.0030 | 0.5110 | 0.4996 | 0.8496 | 0.0541 | 0.4912 | 0.4521 | 0.6603 |
| PO | 0.0541 | 0.6308 | 0.3640 | 0.8811 | 0.0286 | 0.5127 | 0.3170 | 0.7468 | 0.0068 | 0.4424 | 0.0589 | 0.4015 |
| Mismatch | 0.0143 | 0.6304 | 0.9406 | 0.9741 | 0.0030 | 0.5225 | 0.9612 | 0.9194 | 0.0030 | 0.5025 | 0.7525 | 0.7475 |
| LLMU | 0.0541 | 0.6337 | 0.4480 | 0.8865 | 0.0286 | 0.5110 | 0.3058 | 0.7270 | 0.0286 | 0.3296 | 0.0347 | 0.2495 |
| DPO | 0.0541 | 0.6359 | 0.5860 | 0.8852 | 0.0521 | 0.0519 | 0.3437 | 0.7349 | 0.0541 | 0.4264 | 0.0806 | 0.3937 |
| NPO | 0.0068 | 0.6321 | 0.4632 | 0.8950 | 0.0030 | 0.5057 | 0.5196 | 0.8000 | 0.0541 | 0.4788 | 0.4993 | 0.6490 |
| **FLAT** (TV) | 0.0541 | 0.6373 | 0.4391 | 0.8826 | 0.0143 | 0.5168 | 0.4689 | 0.8155 | 0.0068 | 0.5086 | 0.5217 | 0.7067 |
| **FLAT** (KL) | 0.0286 | 0.6393 | 0.5199 | 0.8750 | 0.0143 | 0.5180 | 0.4524 | 0.7850 | 0.0286 | 0.4838 | 0.4942 | 0.6974 |
| **FLAT** (JS) | 0.0541 | 0.6364 | 0.4454 | 0.8864 | 0.0068 | 0.5144 | 0.4572 | 0.8117 | 0.0541 | 0.4959 | 0.4938 | 0.7013 |
| **FLAT** (Pearson) | 0.0541 | 0.6374 | 0.4392 | 0.8857 | 0.0143 | 0.5175 | 0.4591 | 0.8099 | 0.0068 | 0.5093 | 0.5052 | 0.7059 |

performance (FQ Gap), indicating the ineffectiveness of learning the example responses in a naive manner. These results highlight the effectiveness of our method in balancing forget quality and model utility, even without an explicit retaining term in the loss function. We also include additional results using Llama2-7B (Table 9) in Appendix D.1.

## 4.3 ENTITY UNLEARNING

**Experiment Setup.** The TOFU dataset (Maini et al., 2024a) is a synthetic question-answering dataset focused on author biographies, aiming to enable a LLM to unlearn a portion of fictitious authors while retaining knowledge about the rest and real-world facts. The dataset includes 200 fake authors, each with 20 QA pairs, and experiments are conducted with 1%, 5% or 10% of these authors marked for unlearning. We first fine-tuned the target LLM using all dataset to obtain the original LLM. We use Llama2-7B, Phi-1.5B (Li et al., 2023a), and OPT-2.7B as base LLM.

**Evaluation Metrics.** To assess forget quality and model utility, we mainly use two metrics proposed alongside the TOFU dataset, Forget Quality (FQ) and Model Utility (MU) (Maini et al., 2024a). Forget quality, assessed via a p-value from a Kolmogorov-Smirnov test, measures how closely the unlearned model's output matches a model trained only on the retained data in distribution. When the p-value is above 0.01, we say the forgetting is significant. Model utility is the aggregated model performance on held-out retain data regarding fictional authors, real-world author profiles, and world facts. We also report the ROUGE-L score on forget set and retain set. It's important to note that for the forget set, a lower ROUGE-L score does not necessarily indicate better performance. Therefore, we highlight methods where the ROUGE-L score closely matches that of the retained model, as these are considered to produce better results. More metrics can be found in Appendix C.3.1.

**FLAT is always the best in preserving model utility.** As seen in Table 4, it experiences almost no reductions in model utility compared to the original model. On LLaMA2-7B, although GA and KL achieve strong ROUGE-L scores on the retain dataset, their forgetting performance is poor. Similarly, on Phi-1.5B, GD performs well on the retain dataset's ROUGE-L score, but its forgetting performance is insufficient, failing to exceed 0.01.

**FLAT achieves the top two Forget Quality under all three models. FLAT** achieves a ROUGE score that is closest to the Retained LLM on forget dataset under Llama2-7B and Phi-1.5B. PO shows the best forgetting efficiency on Phi-1.5B, but its ROUGE-L scores on both the forget and retain datasets are lower compared to the retained LLM, indicating weaker model utility. A similar issue is observed with GA: while it excels in forgetting performance on the OPT-2.7B model, its model utility remains weaker.

**FLAT achieves the best trade-off.** Our method consistently ranks in the top two across the primary metrics, achieving the best performance in MU. Specifically, KL f-divergence demonstrates strong results in both FQ and MU on LLama2-7B and Phi-1.5B models. Overall, all four f-divergence functions effectively balance forgetting efficiency and model utility. In summary, our method demonstrates the best model utility while achieving top-two results in forgetting performance.

Table 5: Performace on MUSE benchmark using four criteria. We highlight results in **blue** if the unlearning algorithm satisfies the criterion and highlight it in **red** otherwise. For metrics on $D_f$, lower values than the retained LLM are preferred and the lower the better. For metrics on $D_r$, as long as KnowMem is non-zero (indicating retained knowledge), higher values are better. In terms of PrivLeak, the results should be close to 0. Large negative or positive values suggest that they may cause privacy leakage.

| | VerbMem on $D_f$ ($\downarrow$) | | KnowMem on $D_f$ ($\downarrow$) | | KnowMem on $D_r$ ($\uparrow$) | | PrivLeak |
|---|---|---|---|---|---|---|---|
| Original LLM | 58.4 | - | 63.9 | - | 55.2 | - | -99.8 |
| Retained LLM | 20.8 | - | 33.1 | - | 55.0 | - | 0.0 |
| GA | 0.0 | (✔) | 0.0 | (✔) | 0.0 | (✘) | 17.0 |
| KL | 27.4 | (✘) | 50.2 | (✘) | 44.8 | (✔) | -96.1 |
| NPO | 0.0 | (✔) | 0.0 | (✔) | 0.0 | (✘) | 15.0 |
| NPO-RT | 1.2 | (✔) | 54.6 | (✘) | 40.5 | (✔) | 105.8 |
| Task Vector | 56.3 | (✘) | 63.7 | (✘) | 54.6 | (✔) | -99.8 |
| Mismatch | 42.8 | (✘) | 52.6 | (✘) | 45.7 | (✔) | -99.8 |
| GD | 4.9 | (✔) | 27.5 | (✔) | 6.7 | (✔) | 109.4 |
| WHP | 19.7 | (✔) | 21.2 | (✔) | 28.3 | (✔) | 109.6 |
| **FLAT** (TV) | 1.7 | (✔) | 13.6 | (✔) | 31.8 | (✔) | 45.4 |
| **FLAT** (KL) | 0.0 | (✔) | 0.0 | (✔) | 0.0 | (✘) | 58.9 |
| **FLAT** (JS) | 1.9 | (✔) | 36.2 | (✘) | 38.5 | (✔) | 47.1 |
| **FLAT** (Pearson) | 1.6 | (✔) | 0.0 | (✔) | 0.2 | (✔) | 26.8 |

## 4.4 MUSE-NEWS UNLEARNING

**Experiment Setup.** We focus on the task of unlearning on News corpus presented in Shi et al. (2024). News consists of BBC news articles (Li et al., 2023b) collected after August 2023. All articles are randomly divided into forget, retain, and holdout sets. We perform unlearning directly on the pre-trained models provided by the benchmark, following the corresponding experimental setup.

**Evaluation Metrics.** We report the proposed four metrics, *VerbMem* on forget dataset, *KnowMem* on forget and retain dataset, and Privacy leakage (*PrivLeak*). We quantify the verbatim memorization *VerbMem* by prompting the model with the first $l$ tokens from a sequence and comparing the continuation outputted by the model $\theta$ to the true continuation using the ROUGE-L F1 score (Lin, 2004). We gather the model's answers to questions and then average the ROUGE scores for all question-answer pairs in forget dataset or retain dataset to compute the knowledge memorization score *KnowMem*. The *PrivLeak* metric for a good unlearning algorithm should be close to zero, whereas an over/under-unlearning algorithm will get a large positive/negative metric.

**Experiment Results. FLAT** effectively removes verbatim and knowledge memorization of forget dataset and achieve good knowledge memorization of retain dataset. But it can still reveal the membership of $D_f$ in $D_r$. As shown in Table 5, GA, NPO, GD, WHP, and ours perform well in VerbMem and KnowMem on forget dataset, often reduing them even beyond the levels achieved by the retrained model. However, these reductions often come at the cost of significant utility loss on the retain set. Only GD, WHP and ours can perfome good in all memorization related metrics. And **FLAT** (TV) can achieve the lowest VerbMem and KnowMem on $D_f$ and the highest KnowMem on $D_r$ among the three methods. However, none of the methods can achieve satisiable results regarding to the privacy leakage. Since MUSE uses news data, which is highly time-dependent (and thus possibly non-i.i.d.), we advocate for cautious interpretation of the PrivLeak metric (see Appendix C.4.1).

## 4.5 ABLATION STUDIES

**The Effectiveness of Re-weighting Mechanism.** As **FLAT** (KL) demonstrates strong overall performance, we base our ablation study on KL divergence to explore the effectiveness of the implicit re-weighting mechanism within our loss adjustment. This study is conducted on the TOFU dataset using Llama2-7B. For additional results from the ablation studies, please refer to Appendix D.3.

When using preferred template data for unlearning, we compare our method with DPO (without the term $M_{ref}$) and SimPO, as outlined in Appendix C.1. All methods use the same data and have similar formulations, with two terms in the loss function; the only difference lies in the intrinsic re-weighting mechanism. As shown in Table 6, our method achieves the highest number of best

Table 6: Ablation Study of Re-weighting Mechanism on TOFU dataset using Llama2-7B under all metrics. We report ROUGE-L score (R-L), Probability (P), and Truth Ratio (TR) on all four subsets of the TOFU benchmark. Higher scores are better except ROUGE-L and Probability on the Forget Set. The best ones are in **blue**.

| Split | Real Authors | | | Real World | | | Retain Set | | | Forget Set | | |
|---|---|---|---|---|---|---|---|---|---|---|---|---|
| Metric | R-L | P | TR | R-L | P | TR | R-L | P | TR | R-L(↓) | P(↓) | TR |
| Study on Re-weighting Mechanism Using Template IDK Data | | | | | | | | | | | | |
| DPO | **0.9330** | 0.4939 | 0.6384 | 0.8917 | **0.4631** | 0.5646 | **0.8852** | 0.9623 | 0.4407 | 0.5860 | 0.8734 | 0.6240 |
| DPO w/o $M_{ref}$ | 0.9330 | 0.4899 | 0.6333 | 0.8917 | 0.4620 | 0.5642 | 0.8735 | 0.9579 | 0.4388 | **0.4021** | 0.8149 | 0.6326 |
| SimPO | 0.9330 | 0.4902 | 0.6335 | 0.8917 | 0.4624 | **0.5664** | 0.8758 | 0.9577 | 0.4388 | 0.4087 | 0.8128 | **0.6329** |
| **FLAT** (KL) | 0.9180 | **0.4992** | **0.6491** | **0.9060** | 0.4524 | 0.5609 | 0.8750 | **0.9679** | **0.4603** | 0.5199 | **0.7588** | 0.5895 |
| Study on Re-weighting Mechanism Using Retain Data | | | | | | | | | | | | |
| GD | 0.9080 | **0.4728** | **0.6156** | 0.8718 | **0.4439** | **0.5833** | 0.8912 | 0.9657 | **0.4701** | 0.4773 | 0.4238 | 0.5619 |
| **FLAT** (KL)-Retain | **0.9180** | 0.4643 | 0.6099 | **0.8832** | 0.4356 | 0.5690 | **0.9241** | **0.9734** | 0.4697 | **0.4487** | **0.3342** | **0.5879** |

Table 7: Ablation Study of Good Answer Type using three LLMs on TOFU dataset. FQ, MU, R-RL, F-RL represent forget quality, model utility, ROUGE-L on retain dataset and ROUGE-L on forget dataset respectively. The best performance is in **blue**.

| Split | Llama2-7B | | | | Phi-1.5B | | | | OPT-2.7B | | | |
|---|---|---|---|---|---|---|---|---|---|---|---|---|
| Metric | FQ | MU | F-RL(↓) | R-RL | FQ | MU | F-RL(↓) | R-RL | FQ | MU | F-RL(↓) | R-RL |
| Original LLM | 4.4883e-06 | 0.6346 | 0.9851 | 0.9833 | 0.0013 | 0.5184 | 0.9249 | 0.9293 | 0.0013 | 0.5120 | 0.7537 | 0.7494 |
| Retained LLM | 1.0 | 0.6267 | 0.4080 | 0.9833 | 1.0 | 0.5233 | 0.4272 | 0.9269 | 1.0 | 0.5067 | 0.4217 | 0.7669 |
| **FLAT** (TV)-IDK | 0.0541 | 0.6373 | **0.4391** | 0.8826 | 0.0143 | 0.5168 | 0.4689 | 0.8155 | 0.0068 | 0.5086 | 0.5217 | 0.7067 |
| **FLAT** (KL)-IDK | 0.0286 | **0.6393** | 0.5199 | 0.8750 | **0.0143** | **0.5180** | 0.4524 | 0.7850 | 0.0286 | 0.4838 | 0.2212 | 0.4853 |
| **FLAT** (JS)-IDK | 0.0541 | 0.6364 | 0.4454 | 0.8864 | 0.0068 | 0.5144 | 0.4572 | 0.8117 | **0.0541** | 0.4959 | 0.3104 | 0.5658 |
| **FLAT** (Pearson)-IDK | **0.0541** | 0.6374 | 0.4392 | 0.8857 | 0.0143 | 0.5175 | 0.4591 | 0.8099 | 0.0068 | 0.5093 | **0.5052** | 0.7059 |
| **FLAT** (TV)-Normal | 0.0068 | 0.6173 | 0.4941 | 0.9575 | 0.0068 | 0.5104 | 0.4827 | 0.8245 | 0.0030 | 0.5086 | 0.5646 | 0.7355 |
| **FLAT** (KL)-Normal | 0.0068 | 0.6162 | 0.6273 | **0.9719** | 0.0068 | 0.5177 | 0.5377 | **0.8575** | 0.0030 | 0.5082 | 0.5642 | **0.7474** |
| **FLAT** (JS)-Normal | 0.0143 | 0.6178 | 0.4910 | 0.9560 | 0.0013 | 0.5068 | 0.5538 | 0.7313 | 0.0013 | 0.5068 | 0.5538 | 0.7313 |
| **FLAT** (Pearson)-Normal | 0.0068 | 0.6186 | 0.4972 | 0.9546 | 0.0030 | 0.5094 | 0.5554 | 0.7343 | 0.0030 | **0.5094** | 0.5554 | 0.7343 |

results across 12 metrics. When replacing the IDK data with retain data, the results show that the retain version performs better on the Retain Set but worse on Real Authors and Real World compared to **FLAT** (KL). Since GD shares the same data usage and formulation as our method, except for the re-weighting mechanism and utilization of retain data, we compare the retain version to GD. The results show that our method achieves better performance on both the Retain Set and Forget Set, with the decline in Real Authors and Real World performance caused by the use of retain data.

**The Imapct of Good Answer Type.** In the first step of our approach, we intend to generate good example responses for each forget sample. We primarily use the reject-based response "I don't know" (denoted as IDK) as the default choice. In this section, we conduct an ablation study on data usage for **FLAT** to analyze how these good responses impact unlearning performance. Table 7 presents the ablation study of good answer type using three LLMs on TOFU dataset, comparing IDK with random normal responses (denoted as Normal). Table 14 in Appendix D.3 provides the ablation study of different good answer type using Llama2-7B on the HP dataset. Results indicate that using normal responses improves model utility on HP datasets and improves ROUGE-L Score on retain set on TOFU datasets, whereas using IDK responses yields better forgetting quality. Additionally, we observe that the performance across the four divergence functions is relatively similar. KL divergence, in particular, demonstrates more consistent results across the three datasets and three models, likely due to its reduced sensitivity to incorrect or bad answers.

## 5 CONCLUSION

In this paper, we address the limitations of existing LLM unlearning methods, which often rely on the retain data or a reference LLM for response calibration. To overcome these challenges, we propose **FLAT** (Forget data only Loss AdjustmenT), a "flat" loss adjustment approach that eliminates the need for retain data or a reference model. By optimizing the $f$-divergence between the template and forget responses, **FLAT** offers a clear and theoretically grounded solution for balancing forget quality with model utility in LLM unlearning. Through extensive experiments on three key unlearning tasks: copyrighted content unlearning on the Harry Potter dataset, the MUSE benchmark, and entity unlearning on the TOFU dataset, we demonstrate the superior performance of **FLAT**. Our method consistently achieves high unlearning efficiency while preserving overall model utility, showcasing its effectiveness in addressing both practical and theoretical challenges in LLM unlearning.

## ACKNOWLEDGMENT

Y. Wang and Y. Liu are partially supported by the National Science Foundation (NSF) under grants IIS-2143895, IIS-2040800, and IIS-2416896.

## ETHICS STATEMENT

Our proposed approach emphasizes privacy and fairness by addressing potential data privacy concerns during unlearning procedures, particularly with sensitive datasets. We commit to ensuring that no private or proprietary data is mishandled during experiments, and all data used for training and evaluation are publicly available.

## REPRODUCIBILITY STATEMENT

We provide details to reproduce our results in Section 4 and Appendix C, including our experimental setup, evaluation metrics and implementation setting. Additionally, the code and scripts used in our experiments will be made publicly available upon acceptance. Any external libraries or dependencies required to reproduce the results are specified.

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

APPENDIX ARRANGEMENT

The Appendix is organized as follows.

## A   LIMITATIONS AND BROADER IMPACTS

### A.1   BROAD IMPACTS

The proposed **FLAT** method for LLM unlearning has the potential to significantly advance ethical and responsible AI deployment, particularly in addressing key challenges such as privacy concerns, bias, and regulatory compliance. By enabling models to effectively forget specific data without compromising overall model utility, this approach directly addresses issues related to data privacy, including compliance with regulations like GDPR, which mandates data deletion upon user request. The ability to unlearn sensitive or copyrighted information, as demonstrated on datasets like Harry Potter and TOFU, ensures that AI models can be continually refined without propagating harmful or biased content.

Furthermore, the reduced reliance on the retain data or a reference LLM makes **FLAT** more resource-efficient, lowering the computational and financial costs associated with large-scale unlearning. This opens up opportunities for wider adoption across industries and research institutions where access to retain data or additional model resources may be limited or not accessible. The implications of this work span multiple domains, including healthcare, finance, and education, where ethical considerations are paramount.

### A.2   LIMITATIONS

One key limitation of our approach is the unsatisfactory performance in the privacy leakage evaluation on the MUSE dataset. While **FLAT** demonstrates strong unlearning efficiency and retains model utility across several benchmarks, it struggles to prevent privacy leakage. Note that all other tested methods suffer from the same issue, this suggests that further refinement is needed to strengthen the privacy-preserving aspects of existing LLM unlearning approaches. Future work could explore more robust strategies to address privacy leakage while maintaining the balance between unlearning performance and model utility.

## B THEORETICAL ILLUSTRATION AND PROOFS

### B.1 ADDITIONAL EXAMPLES FOR LOSS ADJUSTMENTS UNDER MORE $f$-DIVERGENCE

**Example 2: Jenson-Shannon (JS)** For Jenson-Shannon $f$-divergence, we have $f^*(u) = -\log(2 - e^u), g^*(v) = \log\frac{2}{1 + e^{-v}}$, hence:

$$g^*((\mathbb{P}(x_f, y_e; \theta)) - f^*(g^*((\mathbb{P}(x_f, y_f; \theta))) = \log\frac{2}{1 + e^{-\mathbb{P}(x_f, y_e; \theta)}} - \left[-\log\left(2 - e^{\log\frac{2}{1 + e^{-\mathbb{P}(x_f, y_f; \theta)}}}\right)\right]$$

$$= \log\frac{2}{1 + e^{-\mathbb{P}(x_f, y_e; \theta)}} + \log\left(2 - e^{\log\frac{2}{1 + e^{-\mathbb{P}(x_f, y_f; \theta)}}}\right)$$

$$= \log\frac{2}{1 + e^{-\mathbb{P}(x_f, y_e; \theta)}} + \log\left(2 - \frac{2}{1 + e^{-\mathbb{P}(x_f, y_f; \theta)}}\right)$$

$$= \log\frac{2}{1 + e^{-\mathbb{P}(x_f, y_e; \theta)}} + \log\left(\frac{2e^{-\mathbb{P}(x_f, y_f; \theta)}}{1 + e^{-\mathbb{P}(x_f, y_f; \theta)}}\right)$$

$$= \log\left(\frac{4e^{-\mathbb{P}(x_f, y_f; \theta)}}{(1 + e^{-\mathbb{P}(x_f, y_e; \theta)})(1 + e^{-\mathbb{P}(x_f, y_f; \theta)})}\right).$$

**Example 3: Pearson** For Pearson $f$-divergence, we have $f^*(u) = \frac{u^2}{4} + u, g^*(v) = v$, hence:

$$g^*(\mathbb{P}(x_f, y_e; \theta)) - f^*(g^*((\mathbb{P}(x_f, y_f; \theta))) = \mathbb{P}(x_f, y_e; \theta) - \left(\frac{\mathbb{P}(x_f, y_f; \theta)^2}{4} + \mathbb{P}(x_f, y_f; \theta)\right)$$

$$= -\frac{\mathbb{P}(x_f, y_f; \theta)^2}{4} - \mathbb{P}(x_f, y_f; \theta) + \mathbb{P}(x_f, y_e; \theta).$$

**Example 4: KL** For KL $f$-divergence, we have $f^*(u) = e^{u-1}, g^*(v) = v$, hence:

$$g^*((\mathbb{P}(x_f, y_e; \theta)) - f^*(g^*((\mathbb{P}(x_f, y_f; \theta))) = \mathbb{P}(x_f, y_e; \theta) - e^{\mathbb{P}(x_f, y_f; \theta) - 1}.$$

### B.2 THE DERIVATION OF EMPIRICAL LOSS FUNCTION

According to Eqn. (4), we have:

$$L(x_f, y_e, y_f; \theta) = -\left[\sup_g \left[g(\mathbb{P}(x_f, y_e; \theta)) - f^*(g(\mathbb{P}(x_f, y_f; \theta)))\right]\right]$$

$$= \underbrace{f^*(g^*(\mathbb{P}(x_f, y_f; \theta)))}_{L_f(x_f, y_f; \theta)} \underbrace{-g^*(\mathbb{P}(x_f, y_e; \theta)))}_{L_e(x_f, y_e; \theta)}.$$

Given a dataset $D = \{(x_f^j, y_e^j, y_f^j)\}_{j \in [N]}$, where $y_e$ and $y_f$ are preferred template and original forget responses to the forget prompt $x_f$, we estimate $\mathbb{P}(x_f, y_e; \theta), \mathbb{P}(x_f, y_f; \theta)$ via the following two quantities:

$$\mathbb{P}(x_f, y_e; \theta) := \frac{\sum_{i=1}^{|y_e|} P(\mathcal{M}_\theta(x_f, y_{e,<i}) = y_{e,i})}{|y_e|}, \quad \mathbb{P}(x_f, y_f; \theta) := \frac{\sum_{i=1}^{|y_f|} P(\mathcal{M}_\theta(x_f, y_{f,<i}) = y_{f,i})}{|y_f|}.$$

Given a prompt and the previously generated tokens, $P(\mathcal{M}_\theta(x_f, y_{e,<i}) = y_{e,i})$ and $P(\mathcal{M}_\theta(x_f, y_{f,<i}) = y_{f,i})$ are the probabilities of correctly predicting the next token, where $\mathcal{M}_\theta(x_f, y_{e,<i})$, and $\mathcal{M}_\theta(x_f, y_{f,<i})$ are the predicted token using LLM $\theta$ given input prompt $x_f$ and the already generated tokens $y_{e,<i}$ and $y_{f,<i}$.

Empirically, we can obtain the loss function for the dataset $D$:

$$L_{\textbf{FLAT}}(\theta) = -\mathbb{E}_D\Big[g^*(\mathbb{P}(x_f, y_e; \theta))) - f^*(g^*(\mathbb{P}(x_f, y_f; \theta)))\Big]$$

$$= -\mathbb{E}_D\Big[g^*(\frac{\sum_{i=1}^{|y_e|}\sum_{k=1}^v y_{e,i,k} \cdot h_\theta(x_f, y_{e,<i})_k}{|y_e|}) - f^*(g^*(\frac{\sum_{i=1}^{|y_f|}\sum_{k=1}^v y_{f,i,k} \cdot h_\theta(x_f, y_{f,<i})_k}{|y_f|}))\Big]$$

$$= -\mathbb{E}_D\Big[g^*(\frac{\sum_{i=1}^{|y_e|} h_\theta(x_f, y_{e,<i})}{|y_e|}) - f^*(g^*(\frac{\sum_{i=1}^{|y_f|} h_\theta(x_f, y_{f,<i})}{|y_f|}))\Big].$$

Here, $v$ is the vocabulary size, $y_{e,i,k}$ is the $k$-th element of vector representing the $i$-th token in the good response $y_e$, $y_{f,i,k}$ is the $k$-th element of vector representing the $i$-th token in the forget response $y_f$. Additionally, $h_\theta(x_f, y_{e,<i})_k$ and $h_\theta(x_f, y_{f,<i})_k$ denote the $k$-th entry of the probability distribution for the correctly generated token.

**An example: KL** For KL f-divergence, $f^*(u) = e^{u-1}, g^*(v) = v$, hence, $g^*(\mathbb{P}(x_f, y_e; \theta)) - f^*(g^*(\mathbb{P}(x_f, y_f; \theta))) = \mathbb{P}(x_f, y_e; \theta) - e^{\mathbb{P}(x_f, y_f; \theta)-1}$. We have:

$$L_{\textbf{FLAT}}(\theta) = -\mathbb{E}_D\Big[\frac{\sum_{i=1}^{|y_e|} h_\theta(x_f, y_{e,<i})}{|y_e|} - e^{\frac{\sum_{i=1}^{|y_f|} h_\theta(x_f, y_{f,<i})}{|y_f|}-1}\Big].$$

### B.3 PROOF OF THEOREM 3.4

*Proof.* Remember that we define:

$$\hat{f}_{div}(D_e||D_f) := \mathbb{E}_{Z_e \sim D_e}[\hat{g}(Z_e)] - \mathbb{E}_{Z_f \sim D_f}[f^*(\hat{g}(Z_f))],$$

and

$$f_{div}(\mathcal{D}_e||\mathcal{D}_f) = \sup_g \Big[\mathbb{E}_{\mathcal{Z}_e \sim \mathcal{D}_e}[g(\mathcal{Z}_e)] - \mathbb{E}_{\mathcal{Z}_f \sim \mathcal{D}_f}[f^*(g(\mathcal{Z}_f))]\Big],$$

we first prove the convergence of $\hat{g}$, and then the convergence of $\hat{f}_{div}(D_e||D_f)$.

For the ease of presentation, for any real-valued function $\varrho$, we write $\mathbb{E}_{\mathcal{D}_e}(\varrho) = \mathbb{E}_{z \sim \mathcal{D}_e}[\varrho(z)]$, $\mathbb{E}_{\mathcal{D}_f}(\varrho) = \mathbb{E}_{z \sim \mathcal{D}_f}[\varrho(z)]$, $\mathbb{E}_{D_e}(\varrho) = \mathbb{E}_{z \sim D_e}[\varrho(z)]$, and $\mathbb{E}_{D_f}(\varrho) = \mathbb{E}_{z \sim D_f}[\varrho(z)]$.

Given any $\tilde{g} \in \Phi$, according to Lemma C.1 in (Wei et al., 2021), and the fact that $f^*$ is Lipschitz continuous, we have:

$$||\hat{g} - g^*||^2_{L_2(\mathcal{D}_f)} \precsim [\mathbb{E}_{D_e}[(\hat{g} - g^*)/2] - \mathbb{E}_{\mathcal{D}_e}[(\hat{g} - g^*)/2]]$$
$$- \Big[\mathbb{E}_{D_f}[f^*((\hat{g} - g^*)/2) - f^*(g^*)] - \mathbb{E}_{\mathcal{D}_f}[f^*((\hat{g} - g^*)/2) - f^*(g^*)]\Big]. \quad (5)$$

By using the fact that the true density ratio $Z_e/Z_f$ is bounded below and above, hence, $L_2(\mathcal{D}_e)$ is indeed equivalent to $L_2(\mathcal{D}_f)$. Based on Eqn. (5), Lemma C.2 in (Wei et al., 2021), and the Lipschitz property of $f^*$, with probability at least $1 - c_1 \exp(-N^{r_\Phi/(2+r_\Phi)}/c_1^2)$, we have

$$||\hat{g} - g^*||^2_{L_2(\mathcal{D}_f)} \precsim N^{-1/(r_\Phi+2)}. \quad (6)$$

Note that we have:

$$\Big|\hat{f}_{div}(D_e||D_f) - f_{div}(\mathcal{D}_e||\mathcal{D}_f)\Big|$$
$$\leq |\mathbb{E}_{D_e}[\hat{g} - g^*] - \mathbb{E}_{\mathcal{D}_e}[\hat{g} - g^*]| + |\mathbb{E}_{D_f}[f^*(\hat{g}) - f(g^*)] - \mathbb{E}_{\mathcal{D}_f}[f^*(\hat{g}) - f^*(g^*)]|$$
$$+ |\mathbb{E}_{\mathcal{D}_e}[\hat{g} - g^*] - \mathbb{E}_{\mathcal{D}_f}[f^*(\hat{g}) - f^*(g^*)]| + |\mathbb{E}_{D_e}[g^*] - \mathbb{E}_{\mathcal{D}_e}[g^*]| + |\mathbb{E}_{D_f}[f^*(g^*)] - \mathbb{E}_{\mathcal{D}_f}[f^*(g^*)]|$$
$$= \text{Cons}_1 + \text{Cons}_2 + \text{Cons}_3 + \text{Cons}_4 + \text{Cons}_5. \quad (7)$$

By Lemma C.2 in (Wei et al., 2021), with probability at least $1 - c_1 \exp(-N^{r_\Phi/(2+r_\Phi)}/c_1^2)$, we have

$$\text{Cons}_1 \precsim N^{-2/(r_\Phi+2)}.$$

Similar upper bound holds for $\text{Cons}_2$.

Following from Eqn. (6), with probability at least $1 - c_1 \exp(-N^{r_\Phi/(2+r_\Phi)}/c_1^2)$, we have

$$\text{Cons}_3 \precsim N^{-1/(r_\Phi+2)}.$$

Applying Hoeffding's inequality, with probability at least $1 - c_1 \exp(-N^{r_\Phi/(2+r_\Phi)}/c_1^2)$, we have

$$\text{Cons}_4 \precsim N^{-1/(r_\Phi+2)}.$$

Similar upper bound holds for $\text{Cons}_5$.

Combining the five upper bounds for $\text{Cons}_i$ where $i \in [5]$, with probability at least $1 - c_1 \exp(-N^{r_\Phi/(2+r_\Phi)}/c_1^2)$, we have

$$\left| \hat{f}_{div}(D_e || D_f) - f_{div}(\mathcal{D}_e || \mathcal{D}_f) \right| \precsim N^{-\frac{1}{r_\Phi+2}}.$$

$\square$

## C DETAILED EXPERIMENTAL SETUP

Table 8: Summary of unlearning tasks, including base models, forget datasets, and evaluation metrics.

| Unlearning Task | Base Model | Forget Dataset | Metrics |
|---|---|---|---|
| Copyrighted Content Unlearning | OPT-2.7B, Llama2-7B | Harry Potter Series | BLEU, ROUGE_L, PPL, Zero-shot Acc |
| Entity Unlearning (TOFU) | OPT-2.7B, Llama2-7B, Phi-1.5B | TOFU-Forget01/05/10 | Forget Quality (p-value), ROUGE_L on Forget set, Model Utility |
| MUSE Benchmark | Llama2-7B | BBC News Corps | verbatim and knowledge memorization on $\mathcal{D}_f$, privacy leakage, Utility preservation |

Table 8 summarizes the experimental setups, including base models, forget and retain datasets, and evaluation metrics.

### C.1 FORMULATIONS FOR BASELINE METHODS

In this section, we revisit existing unlearning objectives and unify them under our general loss function framework, as described in Section § 2.2. We provide formulations for GA, GD, KL, and PO, as presented in Maini et al. (2024a), as well as for Mismatch (Liu et al., 2024a), LLMU (Yao et al., 2023), DPO(Rafailov et al., 2024), and NPO (Zhang et al., 2024). Additionally, we include the formulations for DPO without the $M_{ref}$ term and SimPO (Meng et al., 2024), as discussed in Section § 3.4. We also add formulations for Task Vectors (Ilharco et al., 2022) and Who's Harry Potter (WHP)(Eldan & Russinovich, 2023).

**Fine-tuning on retain data** Retraing from scratch is the gold standard for unlearning. HHowever, in real-world scenarios, retain data may not always be available, and retraining a LLM is highly resource-intensive. Alternatively, we can fine-tune the model using retain data for several epochs, which only involves performing gradient descent on $D_r$.

$$L_{\text{Fine-tune}} = \underbrace{\frac{1}{|D_r|} \sum_{(x_r,y_r) \in D_r} \mathcal{L}(x_r, y_r; \theta)}_{\textbf{Retain Loss}}$$

**Gradient ascent (GA)** GA is simple baselines commonly used in traditional machine unlearning settings (Chen et al., 2023; Jia et al., 2023; Fan et al., 2023; Kurmanji et al., 2024). GA reverts the change of the gradient descent during the training with its opposite operation. The rationale of gradient ascent is that a subsequent maximization of prediction loss on the forget dataset $D_f$ would approximately "revert" the optimization on the forget dataset, thus unlearning $D_f$ and approximating a model trained on the retain dataset $D_r$ only.

$$L_{\text{GA}} = \underbrace{-\frac{1}{|D_f|} \sum_{(x_f,y_f) \in D_f} \mathcal{L}(x_f, y_f; \theta)}_{\textbf{Forget Loss}}$$

**Gradient difference (GD)**  Gradient difference has been introduced as simple baseline method in Maini et al. (2024a).It combines fine-tuning and gradient ascent by compute the sum of the two loss terms.

$$L_{\text{GD}} = \underbrace{\frac{1}{|D_r|} \sum_{(x_r,y_r)\in D_r} \mathcal{L}(x_r, y_r; \theta)}_{\textbf{Retain Loss}} - \underbrace{\frac{1}{|D_f|} \sum_{(x_u,y_u)\in D_f} \mathcal{L}(x_f, y_f; \theta)}_{\textbf{Forget Loss}}$$

**KL minimization (KL)**  The KL minimization is adopted from Maini et al. (2024a) and involves a gradient ascent term for forgetting. It also minimizes the Kullback-Leibler (KL) divergence between the predictions on retain data $D_r$ of the reference model (the original model $\theta_o$) and the newly trained model (the unlearned model $\theta$). This term aims to keep the unlearned model's current output distribution on the retain dataset close to its pre-unlearning distribution on the retain samples.

$$L_{\text{KL}} = \underbrace{L_{\text{GA}}}_{\textbf{Forget Loss}} + \underbrace{\frac{1}{|D_r|} \sum_{(x_r,y_r)\in D_r} \sum_{i=1}^{|y_r|} \text{KL}(h_{\theta_0}(x_r, y_{r<i}) \| h_\theta(x_r, y_{r<i}))}_{\textbf{Retain Loss}}$$

**Preference optimization (PO)**  Preference Optimization (PO) differs from the traditional direct preference optimization approach as presented in Rafailov et al. (2024) in that it combines the fine-tuning loss on $D_r$ with a term that teaches the model to respond with 'I don't know' to prompts from $D_f$ (Maini et al., 2024a). Here, $D_{\text{idk}}$ refers to an augmented forget dataset where the model's response to the prompt is 'I don't know.'

$$L_{\text{PO}} = \underbrace{L_{\text{Fine-tune}}}_{\textbf{Retain Loss}} + \underbrace{\frac{1}{|D_{\text{idk}}|} \sum_{x_f, y_{idk}\in D_{\text{idk}}} \mathcal{L}(x_f, y_{idk}; \theta)}_{\textbf{Custom Loss}}$$

Here, the **Custom Loss** utilizes the modified response to the forget prompt to ensure that the model rejects answering questions related to the forget data.

**Mismatch**  Mismatch has the same objective to PO, except it involves constructing a random combination of text sequences $\mathbf{Y}_{\text{rdn}}$. Here, the second term in mismatch is the same as the second term in LLMU (Yao et al., 2023).

$$L_{\text{Mismatch}} = \underbrace{L_{\text{Fine-tune}}}_{\textbf{Retain Loss}} + \underbrace{\sum_{(x_f,\cdot)\in D_f} \frac{1}{|Y_{\text{rdn}}|} \sum_{y_{rdn}\in Y_{rdn}} \mathcal{L}(x_f, y_{rdn}; \theta)}_{\textbf{Custom Loss}}$$

**LLMU (Yao et al., 2023)**  LLMU combines the GA term with two additional terms to learn 1) random completions $Y_{\text{rdn}}$ from $D_{\text{r}}$ (constructed using prompts from $D_f$) to facilitate unlearn and 2) $D_{\text{r}}$ to preserve performance. We use books with similar styles as $D_{\text{r}}$ in our experiments and construct $Y_{\text{rdn}}$ using randomly sampled text sequences from $D_{\text{r}}$.

$$\begin{aligned}
L_{\text{LLMU}} = &- \sum_{(x_f,y_f)\in D_f} \mathcal{L}(x_f, y_f; \theta) \\
&+ \sum_{(x_f,\cdot)\in D_f} \frac{1}{|Y_{rdn}|} \sum_{y_{rdn}\in Y_{rdn}} \mathcal{L}(x_f, y_{rdn}; \theta) \\
&+ \sum_{(x_r,y_r)\in D_r} \sum_{i=1}^{|y_r|} KL(h_{\theta_o}(x_r, y_{r<i}) \| h_\theta(x_r, y_{r<i}))
\end{aligned}$$

We have already unified LLMU in Section § 2.2

**Direct preference optimization (DPO), DPO w/o $M_{ref}$, SimPO**   See Section § 3.4 for more information.

$$L_{\text{DPO},\beta}(\theta) = -\frac{2}{\beta}\mathbb{E}_D\Big[\log\sigma\Big(\underbrace{\beta\log\prod_{i=1}^{|y_e|}h_\theta(x_f,y_{e,<i})}_{\textbf{Custom Loss}} - \underbrace{\beta\log\prod_{i=1}^{|y_f|}h_\theta(x_f,y_{f,<i}))}_{\textbf{Forget Loss}} \quad \underbrace{-M_{ref}}_{\textbf{Retain/ Custom Loss}}\Big)\Big].$$

$$L_{\text{DPO w/o }M_{ref},\beta}(\theta) = -\frac{2}{\beta}\mathbb{E}_D\Big[\log\sigma\Big(\underbrace{\beta\log\prod_{i=1}^{|y_e|}h_\theta(x_f,y_{e,<i})}_{\textbf{Custom Loss}} - \underbrace{\beta\log\prod_{i=1}^{|y_f|}h_\theta(x_f,y_{f,<i}))}_{\textbf{Forget Loss}}\Big)\Big].$$

$$L_{\text{SimPO},\beta}(\theta) = -\frac{2}{\beta}\mathbb{E}_D\Big[\log\sigma\Big(\underbrace{\frac{\beta}{|y_e|}\log\prod_{i=1}^{|y_e|}h_\theta(x_f,y_{e,i})}_{\textbf{Custom Loss}} - \underbrace{\frac{\beta}{|y_f|}\log\prod_{i=1}^{|y_f|}h_\theta(x_f,y_{f,i}))}_{\textbf{Forget Loss}} - \gamma]\Big)\Big]$$

where $\gamma$ is the target reward margin.

**Negative preference optimization (NPO) (Zhang et al., 2024)**   NPO incorporates only the losing response term in DPO (Rafailov et al., 2024), penalizing only the prompt-response pairs in $\mathcal{D}_f$.In the formulation below, $\beta$ represents the inverse-temperature, $\pi_\theta$ is the prediction probability of LLM $\theta$. NPO also has two extended versions that include either the KL term or a fine-tuning term on $\mathcal{D}_r$ to preserve model utility.

$$L_{\text{NPO}} = -\frac{2}{\beta}\mathbb{E}_{D_f}\Big[\log\sigma\Big(-\beta log\frac{\pi_\theta(y_f\mid x_f)}{\pi_{ref}(y_f\mid x_f)}\Big)\Big]$$

$$= -\frac{2}{\beta}\mathbb{E}_{D_f}\Big[\log\sigma\Big(\underbrace{\beta\log\pi_{ref}(y_f\mid x_f)}_{\textbf{Retain/Custom Loss}} - \underbrace{\beta\log\pi_\theta(y_f\mid x_f)}_{\textbf{Forget Loss}}\Big)\Big]$$

$$L_{\text{NPO-KL}} = L_{\text{NPO}} + L_{\text{KL}}$$
$$L_{\text{NPO-RT}} = L_{\text{NPO}} + L_{\text{Fine-tune}}$$

**Task Vectors (Eldan & Russinovich, 2023)**   The taks vector is derived by calculating the weight difference between the original LLM $\theta_o$ and a reinforce LLM $\theta_{reinforce}$, which is the model trained on $D_f$ until it over-fits. This method then subtract this task vector from the original LLM' weights, intuitively moving the model away from the direction it used to adapt to $D_f$. The weights of unlearned model can be obtained as:

$$\theta = \theta_o - (\theta_{reinforce} - \theta_o)$$

**WHP (Eldan & Russinovich, 2023)**   WHP defines the unlearned model $\theta$ as the interpolation between the original model $\theta_o$ and the reinforced model $\theta_{reinforce}$. Let $p_\theta(\cdot|x)$ denote the token distribution parametrized by the model $\theta$ when given a prompt $x$ as input. Then, for any input $x$, WHP samples the next token from:

$$p_\theta(\cdot|x) = p_{\theta_o}(\cdot|x) - \alpha(p_{\theta_{reinforce}}(\cdot|x) - p_{\theta_o}(\cdot|x))$$

where $\alpha$ is a hyperparameter that controls the interpolation between the two models.

## C.2   COPYRIGHTED UNLEARNING ON HP

### C.2.1   EVALUATION METRICS

We use two text similarity metrics to evaluate our models. In each case, the original copyrighted text serves as the reference, and we calculate the similarity between this reference and the text generated by the LLM.

**ROUGE-L**    For the forget dataset, we compute the ROUGE-L recall score (Lin, 2004) between the ground truth responses (the forget responses) and the text generated by the model after unlearning.

**BLEU**    Similarly, we compute the BLEU score (Papineni et al., 2002) for the forget dataset, comparing the ground truth responses to the model's output after unlearning.

A retained model that has never seen the reference text should score low on both metrics, and a successfully unlearned model should perform similarly. Note that for these metrics, values closer to those of the retained model indicate better unlearning, while values that are too large or too small suggest a difference from the retained model Liu et al. (2024a). For Harry Potter datasets, we evaluate similarity using the first 600 generated tokens as per (Jia et al., 2024).

**Perplexity (PPL)**    We assess text fluency and diversity by computing perplexity on the Wikitext (Merity et al., 2016) using the LM Evaluation Harness (Gao et al., 2023). A model with lower perplexity on the fine-tuned data suggests the generated text remains meaningful.

**Zero-shot Accuracy**    We evaluate zero-shot accuracy across various tasks, including BoolQ (Clark et al., 2019), RTE (Dagan et al., 2005), HellaSwag (Zellers et al., 2019), Winogrande (Sakaguchi et al., 2021), ARC-Challenge (Chollet, 2019), ARC-Easy (Chollet, 2019), OpenBookQA (Mihaylov et al., 2018), Piqa (Bisk et al., 2020), and TruthfulQA (Lin et al., 2021). The mean accuracy across these diverse tasks was computed and reported as a comprehensive measure of model utility after unlearning. The higher the average accuracy, the better the results.

### C.2.2    IMPLEMENTATION SETTING.

To demonstrate the copyright removal task, we undertake the fine-tuning of all the models using the complete Harry Potter series. The finetuning procedure for the OPT-2.7B and Llama2-7B models involve a learning rate of 1e-5 and a batch size of 2. AdamW serves as the optimizer for preparing these models. For baseline methods, we set the batch size and learning rate to be the same as in their original papers, and fine-tune for 5 epochs using AdamW optimizer. For our method, we use the same training hyper-parameters as baseline but set the learning rate to be 2e-7.

### C.3    ENTITY UNLEARNING ON TOFU

### C.3.1    EVALUATION METRICS

We utilize the original evaluation metrics designed in the original paper of the TOFU dataset (Maini et al., 2024a).

**Probability**    For each instance in the retain or forget set, we calculate the normalized conditional probability $P(a \mid q)^{1/|a|}$ on the LLM subject to unlearning, where $q$ represents the question, $a$ is the answer, and $|a|$ denotes the number of tokens in the answer. For the real authors and world facts subsets, the dataset provides a set of five answers $\{a_0, \tilde{a}_1, \tilde{a}_2, \tilde{a}_3, \tilde{a}_4\}$, consisting of one correct answer $a_0$ and four perturbed answers that are incorrect. In this case, we compute the ratio $P(a_0 \mid q)^{1/|a_0|} / \sum_{i=1}^{4} P(\tilde{a}_i \mid q)^{1/|\tilde{a}_i|}$.

**Truth ratio**    The truth ratio is computed as the geometric mean of multiple perturbed (incorrect) answers' ($\mathcal{A} = \{\tilde{a}_1, \tilde{a}_2, ...\}$) probabilities over the normalized conditional probability of the paraphrased answer $\hat{a}$.

$$R_{\text{truth}} = \frac{\left( \prod_{i=1}^{|\mathcal{A}|} P(\tilde{a} \mid q)^{|1/\tilde{a}_i|} \right)^{1/|\mathcal{A}|}}{P(\hat{a} \mid q)^{1/|\hat{a}|}}$$

For the real authors and world fact subsets, the original answer $a$ is used in the denominator as no paraphrased answer is available.

**ROUGE-L**    For all subsets of TOFU, we compute the ROUGE-L recall score (Lin, 2004) between the ground truth responses (forget dataset) and the text generated by the model after unlearning.

**Model utility**    The model utility is aggregated as a harmonic mean over nine numbers: the answer probability, truth ratio, and ROUGE recall scores from each of the retain, real authors, and world facts subsets. A higher model utility is always preferred.

**Forget quality**    The forget quality is determined by calculating the p-value from a Kolmogorov-Smirnov (KS) test, which compares two distributions: the truth ratio of the retained model and the truth ratio of the unlearned model on the forget set. A higher p-value suggests that the null hypothesis — that the distributions of the truth ratios from both models are identical — cannot be rejected, indicating that the retained and unlearned models behave similarly.

### C.3.2    IMPLEMENTATION SETTING.

For all LLM unlearning methods, we set the batch size to be 32 following previous works (Maini et al., 2024a; Zhang et al., 2024; Ji et al., 2024) and use consistent learning rates for each model. For Phi-1.5B, we fine-tune the pre-trained models for 5 epochs using learning rate of 2e-5 to obtain the original model. Similarly, we fine-tune Llama2-7B and OPT-2.7B for the same duration with a learning rate of 1e-5. AdamW serve as the optimizer for preparing these models. The unlearning process for all methods, including ours, employs the same learning rate as used during fine-tuning the original models. For all experiments on the TOFU dataset, the training hyperparameters remain consistent across models of the same type.

**Why do we follow the official implementation of TOFU and report the final results, rather than adopting NPO's best-results strategy (Ji et al., 2024; Liu et al., 2024d)?**    It is important to note that the original implementation of TOFU does not evaluate the best result from each epoch but instead use the final model after unlearning to get its evaluations. Also, the baseline methods reported in the TOFU paper reflect the performance of the final model. In contrast, NPO begins to introduce an evaluation strategy that reports the best results achieved at each epoch by their method on the TOFU dataset. **Reporting the best results across all epochs can overstate the model's performance, as it may not fully represent the method's actual unlearning capability.** In real-world scenarios, evaluating during each epoch is often impractical. Instead, it is important to develop a robust method that achieves a good trade-off without time-consuming parameter tuning or requiring frequent evaluations, especially when dealing with larger forget sets. Therefore, to ensure a fair comparison and align with the evaluation settings of the original TOFU paper, we choose to report the final results after unlearning.

## C.4    MUSE-NEWS UNLEARNING

### C.4.1    EVALUATION METRICS

**Note on PrivLeak metric**    The PrivLeak metric used in (Shi et al., 2024) is derived from Min-K% Prob, a membership inference attack method for LLMs. Formally, it is calculated as:

$$\text{PrivLeak} = \frac{\text{AUC}\left(f_{\text{unlearn}}; D_{\text{forget}}, D_{\text{holdout}}\right) - \text{AUC}\left(f_{\text{retrain}}; D_{\text{forget}}, D_{\text{holdout}}\right)}{\text{AUC}\left(f_{\text{retrain}}; D_{\text{forget}}, D_{\text{holdout}}\right)}$$

where the AUC score refers to the standard AUC-ROC score between $D_{\text{forget}}$ and $D_{\text{holdout}}$. While this method indeed discriminates between the forget and holdout distributions as a measure of successful unlearning, it is highly dependent on the data selected for evaluation. Specifically, Min-K% Prob has been shown to yield random-guess accuracy due to modern LLMs being trained on large pretraining corpora for only a small number of iterations, causing fuzzy boundary between members and non-members (Duan et al., 2024).

Furthermore, Maini et al. (2024b) demonstrate that Min-K% Prob results in 1) high variance depending on the random selection of the dataset used for evaluation, 2) better performance when the two subsets (in our case, forget and holdout) are not drawn from the same distribution, and 3) empirically overestimated false positives. The latter finding suggests that the distribution gap (i.e., temporal shift, which is also identified by Duan et al. (2024)) acts as a confounding factor in the discrimination process, since the forget set and holdout set may differ in more than one dimension. Given that MUSE (Shi et al., 2024) uses news data, which is highly time-dependent (and thus possibly non-i.i.d.), we advocate for cautious interpretation of the PrivLeak metric.

Table 9: Unlearning performance of Llama2-7B on the Harry Potter dataset. R-L and Avg. Acc. denote the ROUGE-L score and average zero-shot accuracy across nine LLM benchmarks. We include the original LLM and retained LLM for reference. Some methods, such as PO, LLMU, DPO, and NPO, exhibit strong performance in either forget quality or model utility, but underperform in the other. **FLAT** consistently ranks in the top three in terms of similarity to the retained model, measured by Forget Quality Gap (FQ Gap), while also generating meaningful and diverse outputs, as indicated by perplexity (PPL) and the average zero-shot accuracy (Avg. Acc.). The top three results across the three main metrics are highlighted in **blue**.

| Metric | Forget Quality | | | | | Model Utility | | | |
|---|---|---|---|---|---|---|---|---|---|
| | BLEU($\downarrow$) | BLEU Gap | R-L($\downarrow$) | R-L Gap | **FQ Gap** | **PPL($\downarrow$)** | PPL Gap | **Avg.Acc.** | Acc. Gap |
| Original LLM | 4.0452 | - | 0.1487 | - | - | 8.9524 | - | 0.5617 | - |
| Retained LLM | 0.4903 | - | 0.0442 | - | - | 8.7070 | - | 0.5599 | - |
| GA | 0.0624 | 0.4279 | 0.0134 | 0.0308 | 0.4587 | 47.2769 | -38.5699 | 0.5088 | -0.0511 |
| KL | 0.0976 | 0.3927 | 0.0144 | 0.0298 | 0.4225 | 9.4336 | -0.7266 | 0.5509 | -0.0090 |
| GD | 0.0039 | 0.4864 | 0.0002 | 0.0440 | 0.5304 | 9.1797 | -0.4727 | 0.4902 | -0.0697 |
| PO | 0.0206 | 0.4697 | 0.0015 | 0.0427 | 0.5124 | 8.8364 | -0.1294 | 0.5532 | -0.0067 |
| Mismatch | 0.0670 | 0.4233 | 0.0028 | 0.0414 | 0.4647 | 8.9906 | -0.2836 | 0.5593 | -0.0056 |
| LLMU | 0.3033 | 0.1870 | 0.0317 | 0.0125 | 0.1985 | 9.0530 | -0.3460 | 0.5503 | -0.0096 |
| DPO | 0.7717 | -0.2814 | 0.0552 | -0.0110 | 0.2924 | 8.9597 | -0.2527 | 0.5614 | 0.0015 |
| NPO | 0.9840 | -0.4937 | 0.0656 | -0.0214 | 0.5151 | 9.0397 | -0.3327 | 0.5609 | 0.0010 |
| **FLAT** (TV) | 0.6770 | -0.1867 | 0.0673 | -0.0231 | 0.2098 | 8.9899 | -0.2829 | 0.5592 | -0.0007 |
| **FLAT** (KL) | 0.6829 | -0.1926 | 0.0662 | -0.0220 | 0.2146 | 8.9803 | -0.2733 | 0.5572 | -0.0027 |
| **FLAT** (JS) | 0.6890 | -0.1987 | 0.0684 | -0.0242 | 0.2229 | 8.9910 | -0.2840 | 0.5574 | -0.0025 |
| **FLAT** (Pearson) | 0.6930 | -0.2027 | 0.0680 | -0.0238 | 0.2265 | 8.9906 | -0.2836 | 0.5580 | -0.0019 |

# D EXPERIMENTAL RESULTS

## D.1 COPYRIGHTED UNLEARNING ON HP DATASET

**Performance using Llama2-7B on HP dataset** Table 9 indicates that our method consistently places within the top three across the primary metrics, with TV f-divergence showing the best performance. LLMU achieves the best forgetting effect, comparable to our method, but its model utility is inferior. PO again demonstrates the highest PPL but exhibits poor forgetting performance. While DPO shows good model utility, the difference between our method and DPO in PPL is minimal. Unlike other methods, PO directly leverages fine-tune loss, which helps preserve the model's performance beyond the unlearning. These results highlight the effectiveness of our method, striking a better balance between forgetting quality and model utility.

**Analysis about the edge case of Mismatch** When using OPT-2.7B, the forget quality gap (FQ Gap) between Mismatch and FLAT(TV) is relatively similar. For small LLMs like OPT-2.7B, fine-tuning with the retain data for several epochs can lead to effective forgetting of the forget set. The rationale is that fine-tuning on the forget set may induce catastrophic forgetting over the forget set like continual learning (Parisi et al., 2019). Additionally, OPT-2.7B generally produces lower-quality outputs, which reduces the BLEU gaps between FLAT and Mismatch. As a result, the differences in unlearning performance (FQ Gap) between these two methods appear comparable in this setting.

Mismatch can achieve comparable results using OPT-2.7B on the HP dataset. However, on Llama2-7B, the FQ Gap for the mismatch is 0.4647, and ours is 0.2098 (Table 9). Note that a smaller FQ Gap indicates better unlearning performance. FLAT can show better adoption to different LLMs and different datasets. This might be because the Mismatch fails to keep a good balance between the model utility and the forget quality, while Flat theoretically formulated a reweighting mechanism.

**Parameter Study on Prompt Length for HP dataset** Following Jia et al. (2024); Eldan & Russinovich (2023), we evaluate the forget quality using prompt lengths of 50, 100, 200, 300 on Harry Potter series dataset. Table 10 presents the parameter study of different prompt lengths for assessing forget quality on this dataset. In the main paper, we adopt the prompt length of 200, as suggested by Liu et al. (2024a).

Table 10: Parameter Study of different prompt length on Harry Potter book series dataset. We include the original LLM and retained LLM for reference.

| Split | Prompt Length 50 | | Prompt Length 100 | | Prompt Length 200 | | Prompt Length 300 | |
|---|---|---|---|---|---|---|---|---|
| Metric | BLEU($\downarrow$) | ROUGE-L($\downarrow$) | BLEU($\downarrow$) | ROUGE-L($\downarrow$) | BLEU($\downarrow$) | ROUGE-L($\downarrow$) | BLEU($\downarrow$) | ROUGE-L($\downarrow$) |
| | | | | OPT-2.7B | | | | |
| Original LLM | 3.4492 | 0.1203 | 3.7660 | 0.1273 | 4.1163 | 0.1484 | 3.4924 | 0.1551 |
| Retain LLM | 1.7350 | 0.0944 | 2.3427 | 0.0986 | 2.6072 | 0.1229 | 2.7479 | 0.1261 |
| **FLAT** (TV) | 0.8382 | 0.1090 | 0.9607 | 0.1206 | 1.1955 | 1.4117 | 1.4532 | 0.1628 |
| **FLAT** (KL) | 0.8363 | 0.1107 | 0.9867 | 0.1209 | 1.2743 | 1.3329 | 1.4714 | 0.1657 |
| **FLAT** (JS) | 0.8709 | 0.1101 | 0.9720 | 0.1213 | 1.1986 | 0.1290 | 1.4735 | 0.1635 |
| **FLAT** (Pearson) | 0.8430 | 0.1098 | 0.9624 | 0.1211 | 1.1917 | 1.4155 | 1.4501 | 0.1627 |
| | | | | Llama2-7B | | | | |
| Original LLM | 0.0448 | 0.0049 | 0.3951 | 0.0254 | 4.0452 | 0.1487 | 0.2541 | 0.0275 |
| Retain LLM | 0.0917 | 0.0111 | 0.1664 | 0.0162 | 0.4903 | 0.0442 | 0.2542 | 0.0194 |
| **FLAT** (TV) | 0.0293 | 0.0045 | 0.2627 | 0.0276 | 0.6770 | 0.0673 | 0.2185 | 0.0251 |
| **FLAT** (KL) | 0.0308 | 0.0041 | 0.2592 | 0.0276 | 0.6829 | 0.0662 | 0.2217 | 0.0242 |
| **FLAT** (JS) | 0.0305 | 0.0045 | 0.2512 | 0.0279 | 0.6890 | 0.0684 | 0.2143 | 0.0253 |
| **FLAT** (Pearson) | 0.0310 | 0.0044 | 0.2573 | 0.0281 | 0.6930 | 0.0680 | 0.2163 | 0.0252 |

Table 11: Performance of our method and the baseline methods on TOFU-5% and TOFU-10% dataset using Llama2-7B. FQ, MU, R-RL, F-RL represent forget quality, model utility, ROUGE-L on retain dataset and ROUGE-L on forget dataset respectively. We include the original LLM and retain LLM for reference. The top two results are highlighted in blue.

| Dataset | TOFU-5% | | | | TOFU-10% | | | |
|---|---|---|---|---|---|---|---|---|
| Metric | FQ | MU | F-RL($\downarrow$) | R-RL | FQ | MU | F-RL($\downarrow$) | R-RL |
| Original LLM | 3.0507e-13 | 0.6346 | 0.9918 | 0.9833 | 4.6575e-14 | 0.6346 | 0.9918 | 0.9833 |
| Retained LLM | 1.0 | 0.6281 | 0.3928 | 0.9803 | 1.0 | 0.6225 | 0.3970 | 0.9798 |
| GA | 0.0043 | 0.3545 | 0.2593 | 0.2858 | 2.0608e-13 | 0.0 | 0.0115 | 0.0128 |
| KL | 4.0248e-06 | 0.0538 | 0.0619 | 0.0614 | 1.6347e-10 | 0.0 | 8.3333e-05 | 0.0004 |
| GD | 1.1150e-05 | **0.5532** | **0.3482** | **0.5035** | 2.0608e-13 | 0.0093 | 0.0105 | 0.0336 |
| PO | 3.6025e-09 | 0.2101 | 0.0128 | 0.1385 | 9.1590e-16 | 0.4915 | 0.1091 | **0.6454** |
| Mismatch | 1.8266e-05 | **0.5565** | 0.5470 | **0.7506** | 2.0180e-08 | **0.5106** | 0.6219 | **0.7807** |
| LLMU | 1.1150e-05 | 0 | 0.0142 | 0.0142 | 0.0005 | 0.0 | 0.0112 | 0.0133 |
| DPO | 4.7488e-05 | 0.0 | 0.0167 | 0.0162 | 0.0055 | 0.0 | 0.0147 | 0.0151 |
| NPO | 0.0001 | 0.4630 | 0.3234 | 0.3925 | 0.0017 | 0.3086 | **0.4066** | 0.4383 |
| NPO-RT | 0.0001 | 0.4811 | **0.3331** | 0.4217 | **0.0423** | 0.4093 | **0.4066** | 0.4383 |
| **FLAT** (TV) | **0.0221** | 0.0186 | 0.0047 | 0.0060 | 0.0012 | 0.1624 | 0.0167 | 0.0238 |
| **FLAT** (TV)-RT | **0.1452** | 0.4946 | 0.1991 | 0.3405 | **0.0774** | **0.5204** | **0.3816** | 0.4050 |

## D.2 ENTITY UNLEARNING ON TOFU

**The results on TOFU-5% and TOFU-10%**    Table 11 shows the results on TOFU-5% and TOFU-10% using Llama2-7B. Results indicate that FLAT can achieve a good balance between unlearning efficiency and general language capability.

Note that the retain version of FLAT can achieve the best forget quality on TOFU-5% and TOFU-10% while maintaining high model utility. For the TOFU dataset, which is a synthetic set with separable profiles of 200 authors, using retain data does not significantly blur the boundaries between the forget and retain data. Hence, using the retain data in this task significantly improves performance. However, the primary focus of our work remains on content unlearning (usually only the forget content is known), which reflects more practical and realistic situations encountered in real-world applications.

**TOFU Experimental Results using All Metrics**    Table 12 shows the performance on TOFU using three base LLMs, Llama2-7B, Phi-1.5B, and OPT-2.7 under all metrics. From Table 4, we find that the forget quality on the TOFU-1% is similar to that of the baseline methods may be due to the small size of the forget set (40 samples). When calculating the distributions of truth ratio for such a small sample size, the differences between methods tend to diminish.

**Clarification of the baseline discrepancy**    The difference in forget quality values between ours and NPO reported results arises due to the differences in evaluation settings. We tried our best to evaluate

Table 12: Performance on TOFU dataset using three base LLMs, Llama2-7B, Phi-1.5B, and OPT-2.7 under all metrics. We report ROUGE-L score (R-L), Probability (P), and Truth Ratio (TR) on all four subsets of the TOFU benchmark. Higher scores are better except ROUGE-L and probability on the Forget Set. We include the original LLM and retained LLM for reference. The best two are highlighted in **blue**.

| Split | Real Authors | | | Real World | | | Rerain Set | | | Forget Set | | |
|---|---|---|---|---|---|---|---|---|---|---|---|---|
| Metric | R-L | P | TR | R-L | P | TR | R-L | P | TR | R-L(↓) | P(↓) | TR |
| **Llama2-7B** | | | | | | | | | | | | |
| Original LLM | 0.9350 | 0.4738 | 0.6210 | 0.8846 | 0.4355 | 0.5579 | 0.9833 | 0.9900 | 0.4662 | 0.9851 | 0.9898 | 0.5123 |
| Retained LLM | 0.9230 | 0.4645 | 0.6118 | 0.8932 | 0.4182 | 0.5449 | 0.9833 | 0.9902 | 0.4724 | 0.4080 | 0.1798 | 0.6939 |
| GA | 0.9030 | 0.4754 | 0.6233 | 0.8761 | 0.4432 | **0.5843** | 0.9008 | 0.9546 | **0.4695** | 0.4862 | **0.3566** | 0.5705 |
| KL | **0.9280** | 0.4652 | 0.6092 | 0.8803 | 0.4383 | 0.5691 | **0.9398** | **0.9705** | 0.4655 | 0.5281 | 0.5119 | 0.5626 |
| GD | 0.9080 | 0.4728 | 0.6156 | 0.8718 | 0.4439 | **0.5833** | 0.8912 | 0.9657 | **0.4701** | 0.4773 | 0.4238 | 0.5619 |
| PO | 0.9330 | 0.4850 | 0.6269 | 0.8917 | 0.4582 | 0.5602 | 0.8811 | 0.9627 | 0.4393 | 0.3640 | 0.8695 | **0.6318** |
| LLMU | **0.9330** | 0.4905 | 0.6344 | 0.8917 | **0.4603** | 0.5625 | 0.8865 | 0.9628 | 0.4391 | 0.4480 | 0.8606 | **0.6286** |
| DPO | 0.9330 | **0.4939** | 0.6384 | 0.8917 | **0.4631** | 0.5646 | 0.8852 | 0.9623 | 0.4407 | 0.5860 | 0.8734 | 0.6240 |
| NPO | 0.8930 | 0.4754 | 0.6218 | 0.8746 | 0.4466 | 0.5798 | 0.8950 | 0.9574 | 0.4680 | 0.4632 | **0.3664** | 0.5785 |
| NPO-RT | 0.8830 | 0.4758 | 0.6218 | 0.8746 | 0.4459 | 0.5805 | 0.8958 | 0.9588 | 0.4687 | 0.4519 | 0.3672 | 0.5791 |
| **FLAT** (TV) | 0.9180 | 0.4937 | 0.6459 | 0.8974 | 0.4505 | 0.5591 | 0.8826 | 0.9685 | 0.4607 | **0.4391** | 0.5314 | 0.6026 |
| **FLAT** (KL) | 0.9180 | **0.4992** | **0.6491** | **0.9060** | 0.4524 | 0.5609 | 0.8750 | 0.9679 | 0.4603 | 0.5199 | 0.7588 | 0.5895 |
| **FLAT** (JS) | 0.8980 | 0.4927 | 0.6460 | 0.8974 | 0.4508 | 0.5592 | 0.8864 | **0.9686** | 0.4607 | 0.4454 | 0.5183 | 0.6039 |
| **FLAT** (Pearson) | 0.9180 | 0.4932 | **0.6461** | **0.8974** | 0.4509 | 0.5583 | 0.8857 | 0.9684 | 0.4607 | **0.4392** | 0.5092 | 0.6037 |
| **Phi-1.5B** | | | | | | | | | | | | |
| Original LLM | 0.4073 | 0.3744 | 0.4470 | 0.7503 | 0.4148 | 0.4982 | 0.9199 | 0.9238 | 0.4810 | 0.9607 | 0.9345 | 0.4839 |
| Retained LLM | 0.4240 | 0.3779 | 0.4539 | 0.7585 | 0.4090 | 0.4974 | 0.9269 | 0.9271 | 0.4855 | 0.4272 | 0.1686 | 0.6579 |
| GA | 0.4573 | 0.3638 | 0.4373 | 0.7541 | 0.3978 | 0.4741 | 0.8048 | 0.7748 | 0.4880 | 0.5114 | **0.3268** | 0.5099 |
| KL | 0.4273 | 0.3643 | 0.4370 | 0.7474 | 0.3997 | 0.4764 | 0.8109 | 0.8043 | **0.4889** | 0.5059 | **0.3342** | 0.5091 |
| GD | 0.3907 | 0.3726 | **0.4461** | 0.7605 | 0.4087 | 0.4931 | **0.8496** | **0.8900** | 0.4910 | 0.4996 | 0.4025 | 0.4952 |
| PO | 0.4240 | **0.3728** | 0.4449 | 0.7699 | 0.4190 | **0.5207** | 0.7468 | 0.8747 | 0.4596 | 0.3170 | 0.7362 | **0.5416** |
| LLMU | 0.4240 | 0.3720 | 0.4421 | 0.7785 | **0.4203** | 0.5197 | 0.7270 | 0.8678 | 0.4572 | 0.3058 | 0.7067 | **0.5453** |
| DPO | 0.0420 | 0.3713 | 0.4423 | **0.7785** | 0.4202 | **0.5205** | 0.7349 | 0.8712 | 0.4583 | 0.3437 | 0.6999 | 0.5393 |
| NPO | **0.4573** | 0.3619 | 0.4342 | 0.7417 | 0.3988 | 0.4761 | 0.8000 | 0.7856 | 0.4840 | 0.5196 | 0.3529 | 0.5119 |
| NPO-RT | 0.4473 | 0.3619 | 0.4340 | 0.7474 | 0.3998 | 0.4770 | 0.8024 | 0.7926 | 0.4851 | 0.5193 | 0.3527 | 0.5129 |
| **FLAT** (TV) | 0.4440 | 0.3695 | 0.4390 | **0.7742** | 0.4125 | 0.5040 | **0.8155** | 0.8858 | 0.4709 | **0.4689** | 0.4756 | 0.5395 |
| **FLAT** (KL) | 0.4440 | **0.3735** | **0.4464** | 0.7571 | 0.4175 | 0.5147 | 0.7850 | **0.8874** | 0.4666 | **0.4524** | 0.6285 | 0.5287 |
| **FLAT** (JS) | 0.4340 | 0.3703 | 0.4386 | 0.7588 | 0.4119 | 0.5045 | 0.8117 | 0.8850 | 0.4714 | 0.4572 | 0.4683 | 0.5390 |
| **FLAT** (Pearson) | **0.4540** | 0.3694 | 0.4389 | 0.7674 | 0.4117 | 0.5040 | 0.8099 | 0.8850 | 0.4711 | 0.4591 | 0.4672 | 0.5383 |
| **OPT-2.7B** | | | | | | | | | | | | |
| Original LLM | 0.6687 | 0.3833 | 0.4393 | 0.6433 | 0.3701 | 0.4158 | 0.7494 | 0.8335 | 0.4992 | 0.7537 | 0.8237 | 0.5338 |
| Retained LLM | 0.6487 | 0.3735 | 0.4249 | 0.6278 | 0.3696 | 0.4185 | 0.7669 | 0.8399 | 0.4988 | 0.4217 | 0.1991 | 0.7097 |
| GA | 0.6390 | 0.3774 | 0.4375 | 0.5953 | 0.3644 | 0.4071 | 0.6387 | 0.4097 | **0.4972** | **0.4748** | 0.0722 | 0.6325 |
| KL | 0.6573 | 0.3775 | 0.4346 | **0.6463** | 0.3646 | 0.4071 | 0.6613 | 0.4707 | 0.4958 | 0.4810 | 0.1110 | 0.5902 |
| GD | 0.6453 | 0.3782 | 0.4336 | 0.6084 | 0.3613 | 0.3979 | 0.6603 | 0.6916 | **0.5162** | 0.4521 | **0.1701** | 0.5774 |
| PO | 0.4078 | **0.3874** | **0.4540** | 0.5135 | 0.3705 | **0.4207** | 0.4015 | 0.6922 | 0.4546 | 0.0589 | 0.5220 | 0.6037 |
| LLMU | 0.1528 | 0.3739 | 0.4215 | 0.3946 | 0.3695 | 0.4031 | 0.2495 | 0.6013 | 0.4305 | 0.0347 | 0.3967 | **0.6356** |
| DPO | 0.3478 | 0.3853 | **0.4498** | 0.4915 | **0.3708** | **0.4230** | 0.3937 | 0.6445 | 0.4375 | 0.0806 | 0.3931 | 0.6255 |
| NPO | 0.6573 | 0.3787 | 0.4343 | 0.6281 | 0.3681 | 0.4130 | 0.6490 | 0.4978 | 0.4870 | 0.4993 | **0.1355** | 0.5952 |
| NPO-RT | 0.5698 | 0.3646 | 0.4107 | 0.6264 | 0.3675 | 0.4239 | 0.4620 | 0.2459 | 0.4065 | 0.3627 | 0.1087 | **0.6716** |
| **FLAT** (TV) | **0.6737** | **0.3878** | 0.4457 | 0.6382 | **0.3715** | 0.4137 | **0.7067** | **0.8075** | 0.4858 | 0.5217 | 0.6368 | 0.5601 |
| **FLAT** (KL) | **0.6903** | 0.3802 | 0.4364 | 0.6369 | 0.3672 | 0.4108 | 0.6974 | **0.8078** | 0.4909 | 0.4942 | 0.6735 | 0.5515 |
| **FLAT** (JS) | 0.6723 | 0.3824 | 0.4385 | 0.6369 | 0.3706 | 0.4162 | 0.7013 | 0.7685 | 0.4911 | 0.4938 | 0.5662 | 0.5502 |
| **FLAT** (Pearson) | 0.6737 | 0.3873 | 0.4462 | **0.6467** | 0.3705 | 0.4150 | **0.7059** | 0.8069 | 0.4868 | 0.5052 | 0.6329 | 0.5597 |

the performance of all methods under controlled settings as indicated in TOFU's original paper to ensure a fair comparison. Our implementation is based on the TOFU codebase. The difference between the TOFU official implementation and the NPO implementation is that the NPO evaluates models after every epoch (a total of 10) and reports the epoch with the best forget quality, while the TOFU benchmark uses the final results after five epochs. The difference in reporting policies significantly influences how forget quality is presented and perceived.

## D.3 ABLATION STUDY

**Ablation Study of Reweighting Mechanism on HP dataset** Table 13 demonstrates the effectiveness of the re-weighting mechanism on the Harry Potter dataset. When using the preferred template data for unlearning, our method achieves strong forget quality and comparable model utility. When using retain data, our method outperforms GD, indicating that the re-weighting mechanism improves both unlearning efficiency and model utility.

Table 13: Ablation Study of the Re-weighting Mechanism using Llama2-7B on Harry Potter dataset. R-L and Avg. Acc. denote the ROUGE-L score and average zero-shot accuracy over nine LLM benchmarks. We include the original LLM and retain LLM for reference. The best ones are highlighted in **blue**.

| Metric | Forget Quality | | | | | Model Utility | | | |
|---|---|---|---|---|---|---|---|---|---|
| | BLEU($\downarrow$) | BLEU Gap | R-L($\downarrow$) | R-L Gap | FQ Gap | PPL($\downarrow$) | PPL Gap | Avg.Acc. | Acc. Gap |
| Original LLM | 4.0452 | 3.5549 | 0.1487 | 0.2933 | 3.8482 | 8.9524 | 0.2444 | 0.5617 | 0.0018 |
| Retained LLM | 0.4903 | 0.0 | 0.0442 | 0.0 | 0.0 | 8.7070 | 0.0 | 0.5599 | 0.0 |
| **Study on Re-weighting Mechanism Using Retain Data** | | | | | | | | | |
| GD | 0.0039 | 0.4864 | 0.0002 | 0.0440 | 0.5304 | 9.1797 | -0.4727 | 0.4902 | -0.0697 |
| **FLAT** (KL)-retain | 0.2359 | 0.2544 | 0.0263 | 0.0179 | **0.2714** | **8.9948** | -0.2878 | **0.5591** | -0.0008 |
| **Study on Re-weighting Mechanism Using IDK Data** | | | | | | | | | |
| DPO w/o $M_{ref}$ | 0.7719 | -0.2816 | 0.0523 | -0.0081 | 0.2897 | **8.9674** | -0.2604 | 0.5560 | -0.0039 |
| SimPO | 0.6876 | -0.1973 | 0.0552 | -0.0110 | 0.2723 | 8.9927 | -0.2857 | **0.5593** | -0.0006 |
| **FLAT** (KL) | 0.6829 | -0.1926 | 0.0662 | -0.0220 | **0.2146** | 8.9803 | -0.2733 | 0.5572 | -0.0027 |

Table 14: Ablation Study of the good answer type using Llama2-7B on Harry Potter dataset. R-L and Avg. Acc. denote the ROUGE-L score and average zero-shot accuracy over nine LLM benchmarks. We include the original LLM and retain LLM for reference. The best ones are highlighted in **blue**.

| Metric | Forget Quality | | | | | Model Utility | | | |
|---|---|---|---|---|---|---|---|---|---|
| | BLEU($\downarrow$) | BLEU Gap | R-L($\downarrow$) | R-L Gap | FQ Gap | PPL($\downarrow$) | PPL Gap | Avg.Acc. | Acc. Gap |
| Original LLM | 4.0452 | - | 0.1487 | - | - | 8.9524 | - | 0.5617 | - |
| Retained LLM | 0.4903 | - | 0.0442 | - | - | 8.7070 | - | 0.5599 | - |
| **FLAT** (TV)-IDK | 0.6770 | -0.1867 | 0.0673 | -0.0231 | **0.2098** | 8.9899 | -0.2829 | 0.5592 | -0.0007 |
| **FLAT** (KL)-IDK | 0.6829 | -0.1926 | 0.0662 | -0.0220 | 0.2146 | 8.9803 | -0.2733 | 0.5572 | -0.0027 |
| **FLAT** (JS)-IDK | 0.6890 | -0.1987 | 0.0684 | -0.0242 | 0.2229 | 8.9910 | -0.2840 | 0.5574 | -0.0025 |
| **FLAT** (Pearson)-IDK | 0.6930 | -0.2027 | 0.0680 | -0.0238 | 0.2265 | 8.9906 | -0.2836 | 0.5580 | -0.0019 |
| **FLAT** (TV)-Normal | 0.6942 | -0.2039 | 0.0677 | -0.0235 | 0.2274 | 8.9872 | -0.2802 | 0.5588 | -0.0011 |
| **FLAT** (KL)-Normal | 0.6773 | -0.1870 | 0.0664 | -0.0222 | 0.2092 | **8.9721** | -0.2651 | 0.5580 | -0.0019 |
| **FLAT** (JS)-Normal | 0.6980 | -0.2077 | 0.0678 | -0.0236 | 0.2313 | 8.9888 | -0.2818 | 0.5593 | -0.0006 |
| **FLAT** (Pearson)-Normal | 0.6991 | -0.2088 | 0.0681 | -0.0239 | 0.2327 | 8.9882 | -0.2812 | **0.5594** | -0.0005 |

**Ablation Study of Good Answer Types on HP dataset** Table 14 presents the ablation study on the good answer types using Llama2-7B on the Harry Potter dataset. The results show that using normal responses enhances model utility on the HP dataset, while using IDK responses leads to better forgetting quality.

**Ablation Study of Good Answer Types on TOFU dataset** We designed a prompt instructing GPT-4o not to reveal any information about the two authors included in the forget set from TOFU-1%. Table 15 presents the ablation study on the TOFU-1% dataset using Llama2-7B. Generation is the generated template from GPT-4o. This approach demonstrates the worst performance among the three types, including IDK, normal and generation. One possible explanation is that GPT-4o tends to repeat several words from the question in its answer, which increases its similarity to the ground truth answer and undermines the effectiveness of unlearning.

**Good Answer Generation using GPT-4o.** For TOFU-1% dataset, the system prompt should be following "You are a helpful, respectful, and honest assistant. You must not reply with content that violates copyrights for books, news articles, people personal information, or song lyrics. When I ask questions about the authors Basil Mahfouz Al-Kuwaiti or Nikolai Abilov, you must not provide any information about them or even acknowledge their names since these are the private information. However, for questions about other authors, you should answer accurately and provide relevant information. You should use various and high quality response to answer all my questions even with those about personal information." And then we put each question in forget set into the LLM and get the generated template answer as the good answer.

**Ablation Study of Reweighting Mechanism on TOFU dataset** Table 16 presents the results of the study on the importance of reweighting. The results demonstrate that the reweighting mechanism in FLAT enhances both FQ and MU, achieving an effective balance between unlearning efficiency and overall model capability. The FQ on the TOFU-1% is similar among several baseline methods

Table 15: Ablation Study of good answer type on TOFU-1% dataset using Llama2-7B. Here we mainly focus on FLAT(KL). "Generation" denotes using the generated template from GPT-4o. FQ, MU, R-RL, F-RL represent forget quality, model utility, ROUGE-L on retain dataset and ROUGE-L on forget dataset respectively. We include the original LLM and retain LLM for reference. The best ones are highlighted in **blue**.

| Metric | FQ | MU | F-RL(↓) | R-RL |
|---|---|---|---|---|
| Original LLM | 4.4883e-06 | 0.6346 | 0.9851 | 0.9833 |
| Retained LLM | 1.0 | 0.6267 | 0.4080 | 0.9833 |
| FLAT(KL)-IDK | **0.0286** | **0.6393** | **0.5199** | 0.8750 |
| FLAT(KL)-Normal | 0.0068 | 0.6162 | 0.6273 | 0.9719 |
| FLAT(KL)-Generation | 0.0030 | 0.6338 | 0.9369 | **0.9818** |

Table 16: Ablation Study of the implicit reweighting mechanism on TOFU dataset using Llama2-7B. FQ, MU, R-RL, F-RL represent forget quality, model utility, ROUGE-L on retain dataset and ROUGE-L on forget dataset respectively. We include the original LLM and retain LLM for reference. The best one results are highlighted in **blue**.

| Dataset | TOFU-1% | | | | TOFU-5% | | | | TOFU-10% | | | |
|---|---|---|---|---|---|---|---|---|---|---|---|---|
| Metric | FQ | MU | F-RL(↓) | R-RL | FQ | MU | F-RL(↓) | R-RL | FQ | MU | F-RL(↓) | R-RL |
| Original LLM | 4.4883e-06 | 0.6346 | 0.9851 | 0.9833 | 3.0507e-13 | 0.6346 | 0.9918 | 0.9833 | 4.6576e-14 | 0.6346 | 0.9918 | 0.9833 |
| Retained LLM | 1.0 | 0.6267 | 0.4080 | 0.9833 | 1.0 | 0.6281 | 0.3928 | 0.9803 | 1.0 | 0.6225 | 0.3970 | 0.9798 |
| DPO | **0.0541** | 0.6359 | 0.5860 | **0.8852** | 4.7488e-05 | 0.0 | **0.0167** | **0.0162** | **0.0055** | 0.0 | 0.0147 | 0.0151 |
| SimPO | **0.0541** | 0.6336 | 0.5199 | 0.8750 | 0.0003 | 0.0 | 0.0137 | 0.0151 | 0.0012 | 0.0 | 0.0163 | 0.0158 |
| **FLAT** (TV) | **0.0541** | **0.6373** | **0.4391** | 0.8826 | **0.0221** | **0.0186** | 0.0047 | 0.0060 | 0.0012 | **0.1624** | **0.0167** | **0.0238** |

may be due to the small size of the forget set (40 samples). When calculating the distributions of truth ratio for such size, the differences between methods tend to diminish.

# E  RELATED WORK

## E.1  LLM UNLEARNING

LLM unlearning approaches can be broadly categorized into three families: model-based methods, input-based methods, and data-based methods (Liu et al., 2024c).

**Model-based Methods**   Model-based approaches involve modifying the weights and/or architecture to achieve unlearning. These include gradient ascent (GA) and its variants (Yao et al., 2023; Maini et al., 2024a; Chen & Yang, 2023), as well as model editing techniques (Wu et al., 2023; Ilharco et al., 2022; Belrose et al., 2024). The dominant approach among existing LLM unlearning methods is fine-tuning the original model based on a carefully designed unlearning objective function (Chen & Yang, 2023; Yao et al., 2023; Jia et al., 2024; Li et al., 2024; Yao et al., 2024; Zhang et al., 2024). A common strategy combines forgetting and retaining objectives, applying gradient ascent updates to undesirable data while using regular gradient descent on desirable data (Chen & Yang, 2023; Li et al., 2024). The goal of GA is to maximize the loss on the forget data, essentially reversing the effect of gradient descent during training. Some methods employ custom loss functions that go beyond standard forgetting and retaining losses. For example, Yao et al. (2023) introduce a loss function with three components, where the custom loss reflects advanced techniques or regularization applied to the objectives. Other methods, such as DPO (Rafailov et al., 2024), KTO (Ethayarajh et al., 2024), and NPO (Zhang et al., 2024), utilize reference models to guide the unlearning process.

Chen & Yang (2023) fine-tunes an adapter over the unlearning objective, which acts as an unlearning layer within the LLM. Several works also employ assistant or reinforced LLMs to facilitate unlearning (Ilharco et al., 2022; Eldan & Russinovich, 2023; Huang et al., 2024). Who's Harry Potter (WHP) (Eldan & Russinovich, 2023) is a classic method in LLM unlearning. It involves three components: reinforced training to identify tokens linked to the unlearning target, replacing unique expressions with generic alternatives using the model's predictions, and fine-tuning the model on alternative labels to erase the original text from its memory. Liu et al. (2024d) extends WHP and introduces a causal intervention framework for targeted unlearning. It can achieve strong performance on specific benchmarks (e.g., TOFU) because it relies on targeted input modifications. However, this approach is specifically designed for target unlearning and lacks generalizability and practicality for other tasks. Ji et al. (2024) introduce an assistant LLM that pursues the opposite of the unlearning goals, i.e., remembering forgotten documents and forgetting retained knowledge. The unlearned

LLM is then derived by computing the logit difference between the original and assistant LLMs. UNDIAL (Dong et al., 2024) employs self-distillation to adjust logits, selectively diminishing the impact of targeted tokens. This approach ensures smooth convergence while effectively mitigating catastrophic forgetting.

**Data-based Methods** Data-based methods fine-tune the LLM using a set of modified responses. This approach often begins by generating altered outputs (e.g., refusal-based responses), such as obliterated responses (Choi et al., 2024), inverted facts (Gu et al., 2024), or in-domain plausible alternatives (Mekala et al., 2024). These generated responses are then used to guide the unlearning process. Mekala et al. (2024) propose Alternate Preference Optimization, which utilizes in-domain positive feedback on the forget set, complementing the usual negative feedback to overcome the limitations of relying solely on negative feedback during unlearning. In this work, we employ reject-based template outputs as the modified "good" responses for the forgotten samples.

**Input-based Methods** Input-based methods craft input instructions (Pawelczyk et al., 2023; Muresanu et al., 2024; Thaker et al., 2024; Bhaila et al., 2024; Gao et al., 2024; Liu et al., 2024a), such as in-context examples and prompts, to steer the original LLM toward the unlearning objective without altering the model's parameters. These approaches aim to achieve unlearning in the output space rather than in the parameter space. Among these methods, a notable baseline by Liu et al. (2024a) uses an external prompt classifier as a guardrail, applying embedding corruptions to the identified prompts. The authors demonstrate that this corruption scheme results in distribution-wise similarity to the retrained model.

In this work, we propose a novel loss adjustment method for LLM unlearning, which simultaneously utilizes available example responses, effectively combining data-based and model-based methods.

## E.2 MACHINE UNLEARNING

In response to the data regulation requirements (Hoofnagle et al., 2019), machine unlearning (MU) has emerged as a critical process to remove the influence of specific data points, data classes, or even higher-level data concepts from a trained machine-learning model. One direct unlearning method involves retraining the model from scratch after removing the forgotten data from the original dataset, which is often considered the gold standard (Liu et al., 2024b; Fan et al., 2024). However, this approach comes with significant computational demands. To alleviate this, most research focuses on developing approximate but much faster unlearning techniques, including gradient ascent (Thudi et al., 2022; Graves et al., 2021), influence unlearning (Izzo et al., 2021; Warnecke et al., 2021; Wu et al., 2022), Fisher forgetting (Becker & Liebig, 2022; Golatkar et al., 2020), finetuning-based approaches (Liu et al., 2024b; Fan et al., 2023), and loss correction-related unlearning (Adolphs et al., 2022; Wang et al., 2023; Di et al., 2024).

