# OpenReview forum: "LLM Unlearning via Loss Adjustment with Only Forget Data"
_ICLR.cc/2025/Conference — ICLR 2025 Poster_

### Official Review · Reviewer_DTQM · 2024-11-03

**Soundness:** 4
**Presentation:** 3
**Contribution:** 4
**Rating:** 6
**Confidence:** 4

**Summary:**

This paper proposes FLAT, an LLM unlearning method that only requires "forget data" for parameter update. FLAT is driven by an optimization loss that maximizes the f-divergence between the forget data and the forget query paired with expected unlearned responses. The authors finds in this way model can implicitly choose a balance between promoting exemplary and suppressing bad generations of forget data. Experimental results show FLAT's superiority against previous baselines. More importantly, FLAT does not require optimizing models on "retain data" or any reference model to achieve unlearning.

**Strengths:**

1. FLAT is a method without using "retain data" or reference models, which is novel and potentially more efficient against previous baselines.
2. Paper is easy to follow. Proof is provided.
3. Experiments are comprehensive and convincing.
4. I like the informative appendices.

**Weaknesses:**

1. Not really a lot of weaknesses. I just feel that some experimental settings should be more clearly discussed. Please refer to the questions below.
2. Some repeated contents in the main body and I suspect some equation is not correct. But I can understand. Please refer to the questions.

Please address my questions one by one. Good luck with the rebuttal.

**Questions:**

1. Should line 235 be log prob inthe equation? Since you have a sum rather than multiplication on probabilities.
2. Line 371. PO should have the lowest PPL as shown in Table 3.
3. On TOFU dataset. Did you first finetune base LLMs on fictitious data (including forget, reatin and held-out) and then perform unlearning? I think you should add one sentence explaining that. And the forget quality measures the how close the unlearned model’s output matches a model trained only on the retain data. Should here the input into the unlearned model the input of forget data? Then why we should expect the unlearned model resembles a model trained only on the retain data? I feel very confused with this setting. Could you provide clearer explanation?
4. I am also wondering why on Harry Potter you used Forget Quality Gap, which is the difference in ROGUE L and and BLEU. However in TOFU you choose to directly compare the ROGUE-L between unlearned model and retained model. As pointed our in line 408, low ROGUE-L doesn’t necessarily indicate to better performance. Then why you still use Forget Quality Gap in Harry Potter?
5. Is line 512-515 repeated content?

**Details Of Ethics Concerns:**

I believe LLM unlearning requires ethical review. In this submission, some dataset like Harry Potter refers to copyrighted data. And some dataset like like MUSE is related to privacy leakage and potentially fairness and bias. But I want to point out that the authors develop unlearning method to avoid harmfulness.

---

> ### Author Response · Authors · 2024-11-22
>
> Thanks so much for your positive and valuable review. Your encouraging comments and thoughtful feedback are highly motivating and will help us further refine our paper.
>
> **Q1:Should line 235 be log prob in the equation?**
>
> The two probabilities mentioned in line 235 represent the empirical estimation of the data distribution, as indicated in Eq. 3, which aims to maximize the f-divergence between the good- and bad-sample distributions. In empirical cases, we do not employ the cross-entropy loss. Instead, FLAT formalizes the loss within the f-divergence framework to estimate the data distributions. The probabilities serve as the proxies for the good and bad sample distributions, aligning with the theoretical foundations [1].
>
> **Q2:Line 371. PO should have the lowest PPL as shown in Table 3**
>
> Thank you for catching that. Yes, it should indeed refer to the lowest PPL. We appreciate your attention to detail.
>
> **Q3: The evaluation setting**
>
> Thank you for raising these important questions. On the TOFU dataset, we followed the instructions of TOFU and fine-tuned the base LLM using all data. We added the sentence in the revision.
>
> **Explanation for unlearning goal.** In our method, the input to the unlearned model consists only of the forget data and templated reject-based answers. The ideal unlearning solution is to retrain the model from scratch using only the retain data after removing specific training data points [2]. However, retraining LLMs is expensive due to high computational costs and the need to access the entire training dataset. To address these challenges, existing literature focuses on developing approximate unlearning methods to make the unlearned model resemble the retained LLM as much as possible.
>
> Given that the original model contains all prior knowledge, the unlearning process should enable the model to retain its performance on the non-forget data while erasing any information related to the forget set.  Ideally, the unlearned model's behavior should match that of a model trained solely on the retain data, ensuring that no information from the forget set is retained.
>
> **Q4: Questions related to forget quality metrics**
>
> For the Harry Potter dataset, we evaluate Forget Quality (FQ) using ROUGE-L and BLEU scores, following previous work [3]. Since lower scores don’t always indicate better performance, we calculate the Forget Quality Gap (FQ Gap), which is the sum of the BLEU Gap and ROUGE-L Gap, as done in [4]. These gaps are the absolute difference between the retained model and the unlearned model. The smaller FQ Gap towards the retained model should represent better unlearning performance.
>
> For the TOFU dataset, we adopt the metric proposed in the original paper, reporting ROUGE-L scores on both the forget set and retain set. Similarly, a lower ROUGE-L score on the forget set does not necessarily signify better unlearning. Therefore, we highlight methods where the ROUGE-L score closely matches that of the retained model, as these are considered to produce better results.
>
> **Q5: Is line 512-515 repeated content?**:
>
> Thank you for pointing that out. We apologize for the oversight. The updated version will address these minor issues and ensure correctness.
>
> Please let us know if you have any more questions! Thank you!
>
> [1] When optimizing $ f $-divergence is robust with label noise.
>
> [2] Rethinking machine unlearning for large language models.
>
> [3] SOUL: Unlocking the Power of Second-Order Optimization for LLM Unlearning
>
> [4] Large Language Model Unlearning via Embedding-Corrupted Prompts

---

> > ### Comment · Reviewer_DTQM · 2024-11-23
> > **Good Paper**
> >
> > I think I have got better impression after considering authors' rebuttal. I will keep my scores for now and I confirm my judgment is certain. I would like to give 7 for overall assessment but we don't have it this time. In short, I wish it to appear at ICLR 2025.

---

> ### Author Response · Authors · 2024-11-25
> **Thanks for the Positive Feedback**
>
> Thank you for taking the time to carefully consider our rebuttal and for your thoughtful feedback. We are grateful for your positive impression and support for our work. Your encouraging comments and wish to see our paper at ICLR 2025 mean a lot to us, and we deeply appreciate your recognition of our efforts.
>
> Thank you so much! Wish you all the best!

---

### Official Review · Reviewer_vnkm · 2024-11-04

**Soundness:** 2
**Presentation:** 2
**Contribution:** 2
**Rating:** 6
**Confidence:** 2

**Summary:**

This paper explores a method for LLM unlearning with only the data designated for forgetting, rather than relying on both forget and retain data. The proposed method employs a weighted loss function comprising two terms: one term encourages the model to produce template/refusal response for the forget data, and the other term discourages the model from generating the original. The experimental results suggest that this method achieves a favorable trade-off between forgetting accuracy and retaining the model's overall performance.

**Strengths:**

* The paper studies a relaxed unlearning setting for LLMs that does not require retain data, which may be challenging to obtain in some cases.
* The experiment involves multiple benchmarks, diverse LLMs, and various baseline unlearning methods to offer a comprehensive assessment of the proposed approach.

**Weaknesses:**

* Unclear Motivation for f-divergence Reweighting: I did not quite follow the method section. Specifically, I think using f-divergence to reweight the loss terms with coefficients $\lambda_e$ and $\lambda_f$ remains unclear. The method section (specifically lines 197-215, step 3) states that this maximizes the f-divergence between $D_e$ and $D_f$. But I'm not fully clear why this is beneficial for LLM unlearning.
* Limited Performance improvement: The experimental results do not clearly demonstrate a significant advantage of the proposed method over baseline methods. For example, the forget quality on the TOFU dataset is similar to that of the baseline methods. Additionally, the NPO method paper reports a substantially higher forget quality (close to 1.0) in the TOFU-1% unlearning scenario, yet Table 4 reports a much lower forget quality for NPO (6e-3).
* Scalability Concerns: The experiments focus only on the TOFU-1% unlearning setting for TOFU dataset, which involves forgetting information on only 2 authors out of 200 in the synthetic dataset. There is no result on the performance of the method with larger forget sets, such as the TOFU-5% or TOFU-10% settings, raising concerns about the scalability and generalizability of the proposed method.


Please let me know if I have misunderstood any points in the paper.

**Questions:**

* How is the alternative answer sampled? Is it generated by the model undergoing unlearning, or by a separate LLM?
* It seems that the primary difference between the proposed objective function and the SimPO objective function is the weight for the two loss terms. Is there an intuitive explanation for why the proposed f-divergence-based weighting would perform better than the approach used in SimPO?

---

> ### Author Response · Authors · 2024-11-22
> **Response (1/3) W1 and W3**
>
> We sincerely appreciate the reviewer’s time and effort in reading our paper and offering thoughtful suggestions and constructive feedback.
>
> **W1: Unclear Motivation for f-divergence Reweighting.**
>
> **f-divergence to reweight two loss terms:** Eq. 2 consists of two terms. The first term encourages the model to learn the exemplar good answer through gradient descent, while the second term drives the model away from the bad answer (forget data) via gradient ascent. However, determining the appropriate weighting for these terms is non-trivial, as it requires balancing the importance of learning good answers and forgetting bad answers. This is where f-divergence comes into play. The f-divergence provides a rich family of divergence functions that help maximize the model's behavior on two different distributions (good vs bad). Its variational form yields an easy-to-optimize objective function that has an interpretable structure: first term encourages the model to align with the good data distribution by maximizing the probability of exemplar answers, while second term penalizes overlap with the bad data distribution by minimizing the probability of forget data answers.
>
> **Maximize the f-divergence:** Eq. 3 represents the variational form of the f-divergence between the good and bad data distributions. The empirical estimation of f-divergence, as described in Eq. 4, is constructed using the generated data distributions and can serve as the loss function when finetuning the LLM. This estimation provides a natural reweighting mechanism that closely approximates the true f-divergence, effectively modeling various data distributions following from [1,2]. Theorem 3.4 supports this by showing that the empirical alternative achieves the optimal non-parametric rate of convergence toward the true f-divergence. This ensures that the estimated reweighting is robust and well-aligned with the underlying data distributions.
>
> **Benefits for LLM unlearning:** We use f-divergence to provide a principled approach for reweighting the two loss terms. Specifically, maximizing the f-divergence between the data distributions of good (template reject-based) and bad (forget) answers ensures that the model effectively separates these two distributions. This separation directly aligns with the goals of unlearning: to retain useful knowledge while forgetting undesired information. It ensures that the model learns the correct template pattern for the questions in the forget set (e.g., reject-based answers) while effectively forgetting the original answers from the forget set without compromising the performance of retained knowledge. FLAT adjusts the LLM to increase the probability of generating preferred reject-based answers.
>
> **W3: Scalability Concerns:**
>
> We have added experiments on TOFU-5% and TOFU-10% using Llama2-7B. The results demonstrate that FLAT consistently ranks in the top two for FQ. For the complete results, please refer to Table 12 in the revision.
>
> Table 2 in this rebuttal presents part of the results for TOFU-5% and TOFU-10% using Llama2-7B. Note that we only report the results on the final models that are consistent with the TOFU implementation. This differs from the NPO implementation, which evaluates models after every epoch (a total of 10) and reports the epoch with the best FQ. We also report the version using the retain data as the good answer.
>
> Table 2 For both metrics, FQ and MU, higher values indicate better performance.
>
> | Model         | TOFU-5% FQ | TOFU-5% MU | TOFU-10% FQ | TOFU-10% MU |
> |---------------|------------|------------|-------------|-------------|
> | GA            | 0.0043     | 0.3545     | 2.0608e-13  | 0.0000      |
> | PO            | 3.6025e-09 | 0.2101     | 9.1590e-16  | 0.4915      |
> | NPO           | 0.0001     | 0.4630     | 0.0017      | 0.3086      |
> | NPO-RT        | 0.0001     | 0.4811     | 0.0423      | 0.4093      |
> | FLAT (TV)     | 0.0221     | 0.0186     | 0.0012      | 0.1624      |
> | FLAT (TV)-RT  | **0.1452** | **0.4946** | **0.0774**  | **0.5204**  |

---

> > ### Author Response · Authors · 2024-11-22
> > **Response (2/3) W2**
> >
> > **W2: Limited Performance improvement.**
> >
> > **FLAT demonstrates consistently competitive performance while requiring less data, striking a good balance between unlearning efficiency and general language capability.** Unlike some baselines that utilize the retain data for fine-tuning or reference models, FLAT relies solely on the forget data, making it particularly suited for real-world scenarios with resource constraints. While it may not consistently outperform all baselines leveraging retained data or reference models across every dataset and model, FLAT remains a competitive and practical choice with broad applicability. For comparison, we also report a version of FLAT using the retain data as the good answer, following the NPO paper.
> >
> > **FLAT can achieve the top two best methods across three datasets.** FLAT is consistently ranked as the top two across unlearning efficiency and model ability and achieves a strong balance in the Harry Potter dataset. For MUSE, our method achieves the best forget quality and the highest model utility among all methods that satisfy the good trade-off criterion. As for the experiemnt on the TOFU dataset, FLAT achieves the best MU and ranks in the top two for FQ across all three models. In TOFU-5% and TOFU-10%, FLAT ranks among the top three methods for Forgetting Quality (FQ), with a significant gap between FLAT and other baselines in FQ, as highlighted in Table 2 in this rebuttal.
> >
> > The forget quality on the TOFU-1% is similar to that of the baseline methods may be due to the small size of the forget set (40 samples). When calculating the distributions of truth ratio for such a small sample size, the differences between methods tend to diminish.
> >
> > **Note that the retain version of FLAT can achieve the best forget quality on TOFU-5\% and TOFU-10\% while maintaining high model utility.** For the TOFU dataset, which is a synthetic set with separable profiles of 200 authors, using retain data does not significantly blur the boundaries between the forget and retain data. Hence, using the retain data in this task significantly improves performance. However, the primary focus of our work remains on content unlearning (usually only the forget content is known), which reflects more practical and realistic situations encountered in real-world applications.
> >
> > **The difference in forget quality values between ours and NPO reported results arises due to the differences in evaluation settings.** We tried our best to evaluate the performance of all methods under controlled settings as indicated in TOFU’s original paper to ensure a fair comparison. The difference between the TOFU official implementation and the NPO implementation is that the NPO evaluates models after every epoch (a total of 10) and reports the epoch with the best forget quality, while the TOFU benchmark uses the results after five epochs. The difference in reporting policies significantly influences how forget quality is presented and perceived.
> >
> > Table 1 in this rebuttal provides the best results of each epoch across different methods. Using the implementation of NPO, we set lr=1e-5, epoch=10, evaluate at each epoch, and report the best performance. Under this setting, even a simple algorithm like GA can achieve relatively good FQ (the reported FQ for GA on TOFU-5% is under 0.1 in NPO paper, while here it is 0.2404). This indicates that **reporting the best results across all epochs can overstate the model’s performance, as it may not fully represent the method's actual unlearning capability.**
> >
> > Table 1 The **best results of each epoch** on TOFU dataset using Llama2-7b under 1%, 5%, 10% settings. For both metrics, FQ and MU, higher values indicate better performance.
> >
> > | Model | TOFU-1% FQ | TOFU-1% MU | TOFU-5% FQ | TOFU-5% MU | TOFU-10% FQ | TOFU-10% MU |
> > |-------|------------|------------|------------|------------|-------------|-------------|
> > | GA    | **0.9900** | 0.5215     | 0.2404     | 0.0134     | **0.5824**  | **0.5119**  |
> > | NPO   | 0.9188     | 0.5209     | 0.7431     | 0.4216     | 0.0996      | 0.3086      |
> > | FLAT  | 0.9188     | 0.5142     | **0.7894** | **0.5019** | 0.1323      | 0.5024      |

---

> > > ### Author Response · Authors · 2024-11-22
> > > **Response (3/3) Q1 and Q2**
> > >
> > > **Q1: How is the alternative answer sampled? Is it generated by the model undergoing unlearning or by a separate LLM?**
> > >
> > > In our work, the alternative answers are not generated by any LLM. Instead, we use the template reject-based answers provided in the TOFU paper. These template answers include responses like "I do not know the answer" (or any one of 100 versions of this response). The ablation studies in Table 7 and Table 14 explore the impact of using two different types of alternative answers. While generating alternative answers using an LLM could be a promising method, it represents a separate line of research in LLM unlearning (Data-based method) and could serve as a valuable direction for follow-up studies.
> > >
> > > **Q2: The difference between SimPO and FLAT.**
> > >
> > > While the proposed objective function and SimPO share similarities in their formulation, they differ in their underlying principles, the weighting of the two loss terms, and the activation functions applied to each term.
> > >
> > > SimPO is to directly optimize the model toward good answers,  which assigns equal weights to the two loss terms. Unlike SimPO, our method dynamically adjusts the weights, allowing for greater flexibility in balancing the importance of forgetting bad answers and learning good answers. Our method provides a theoretical guarantee for estimating the weights of the two loss terms using the f-divergence perspective. This eliminates the need for (clueless) manual tuning and ensures that the weights are optimized to achieve the best trade-off between forget quality and model utility. This is particularly important in scenarios where it is difficult to determine which term should contribute more.
> > >
> > > In addition to the weights, the activation functions for the two loss terms also differ. The proposed method introduces a reweighting mechanism not only between the two loss terms (inter-term reweighting) but also within each term itself (intra-term reweighting). This approach ensures that each term contributes appropriately to the overall optimization objective, leading to better results than SimPO, which uses a simpler, uniform weighting scheme.
> > >
> > > Our method is designed to achieve a stable balance between learning and forgetting. By using f-divergence as the guiding principle, the optimization process becomes more robust to dataset variations, ensuring consistent performance across different settings. This stability makes it easier to maintain good forget quality while preserving model utility. Additionally, the intra-term reweighting further enhances the flexibility and adaptability of FLAT, making it better suited for the complex objectives of LLM unlearning.
> > >
> > > **Please let us know if you have any more questions! Thank you!**
> > >
> > > [1]  f-GAN: Training Generative Neural Samplers using Variational Divergence Minimization
> > >
> > > [2] When optimizing $ f $-divergence is robust with label noise.

---

> ### Comment · Reviewer_vnkm · 2024-11-23
>
> Thank you for the detailed response. However, I still have two concerns:
>
> 1. Proposed Weighting: I’m still not convinced of its utility. As mentioned in the rebuttal, the two-term loss aims to increase the probability of an alternative answer but decrease the original answer probability, which aligns closely with the DPO loss that is widely used in LLM unlearning. However, the performance comparison does not show a clear advantage for the proposed reweighting on the TOFU dataset. And the performance reported has some discrepancies with previous works, which raises a big concern for me.
>
> 2. Baseline Discrepancy: The reported performance for baseline, such as the NPO, differs from prior works. For example, both [1] and [2] report ~1e-1 FQ for TOFU-10% with NPO and NPO-RT, consistent with the NPO paper, but the rebuttal results has some discrepancy.
>
> [1] Reversing the Forget-Retain Objectives: An Efficient LLM Unlearning Framework from Logit Difference
> [2] Simplicity Prevails: Rethinking Negative Preference Optimization for LLM Unlearning

---

> > ### Author Response · Authors · 2024-11-25
> > **Response (1/3)  Ablation Study on the Proposed Implicit Reweighting Mechanism**
> >
> > **Concern 1:Proposed Weighting**
> >
> > In Table 12 in the revision, we provide the whole results on TOFU-5\% and TOFU-10\%. And FLAT ranks among the top three methods for FQ. **And the retain version of FLAT can achieve the best FQ on TOFU-5\% and TOFU-10\% while maintaining high model utility.**  FLAT consistently demonstrates competitive performance when compared to existing methods such as DPO on Harry Potter and MUSE.
> >
> > Our method provides a theoretical guarantee for estimating the weights of the two loss terms using the f-divergence perspective. This eliminates the need for (clueless) manual tuning and ensures that the weights are optimized to achieve the best trade-off between forget quality and model utility. This is particularly important in scenarios where it is difficult to determine which term should contribute more.
> >
> > Table 14 in the revision presents an ablation study of the reweighting mechanism on the HP dataset. When using similar data (forget and IDK template answers), **the FQ Gap for SimPO is 0.2723, whereas FLAT is reduced to 0.2146.**
> >
> > Below, we present the results of the study on the importance of reweighting. **The results demonstrate that the reweighting mechanism in FLAT enhances both FQ and MU, achieving an effective balance between unlearning efficiency and overall model capability.** The FQ on the TOFU-1\% is similar among several baseline methods may be due to the small size of the forget set (40 samples). When calculating the distributions of truth ratio for such size, the differences between methods tend to diminish.
> >
> > Table 3 Ablation Study of the reweighting mechanism on TOFU dataset under 1\%, 5\%, and 10\%.
> > | Model    | TOFU-1% FQ | TOFU-1% MU | TOFU-5% FQ   | TOFU-5% MU | TOFU-10% FQ | TOFU-10% MU |
> > |----------|------------|------------|--------------|------------|-------------|-------------|
> > | DPO      | **0.0541** | 0.6359     | 4.7488e-5    | 0.0        | **0.0055**  | 0.0         |
> > | SimPO    | **0.0541** | 0.6336     | 0.0003       | 0.0        | 0.0012      | 0.0         |
> > | FLAT(TV) | **0.0541** | **0.6373** | **0.0221**   | **0.0186** | 0.0012      | **0.1624**  |
> >
> > TOFU-1% dataset, FLAT achieves the highest number of best results across 12 metrics.
> > | Model       | R-L (Real Authors) | P (Real Authors) | TR (Real Authors) | R-L (Real World) | P (Real World) | TR (Real World) | R-L (Retain Set) | P (Retain Set) | TR (Retain Set) | R-L (Forget Set) $\downarrow$ | P (Forget Set) $\downarrow$ | TR (Forget Set) |
> > |-------------|---------------------|------------------|-------------------|------------------|----------------|-----------------|------------------|----------------|-----------------|------------------|----------------|-----------------|
> > | SimPO       | **0.9930**  | 0.4902 | **0.6491**  | **0.9060**   | **0.4524**    | **0.5609** | 0.8750  | 0.9679 | 0.4603  | 0.5199  | 0.7588   | 0.5895  |
> > | FLAT(TV)    | 0.9180     | **0.4937**       | 0.6459   | 0.8974   | 0.4505     | 0.5591      | **0.8826**  | **0.9685**     | **0.4607**      | **0.4391**       | **0.5314**     | **0.6026** |
> >
> > TOFU-5% dataset
> > | Model       | R-L (Real Authors) | P (Real Authors) | TR (Real Authors) | R-L (Real World) | P (Real World) | TR (Real World) | R-L (Retain Set) | P (Retain Set) | TR (Retain Set) | R-L (Forget Set)$\downarrow$  | P (Forget Set)$\downarrow$  | TR (Forget Set) |
> > |-------------|---------------------|------------------|-------------------|------------------|----------------|-----------------|------------------|----------------|-----------------|------------------|----------------|-----------------|
> > | SimPO       | 0.0053             | 0.4200           | 0.5404           | 0.0    | 0.4284      | 0.5345    | **0.0151**       | **0.5050**     | **0.3619**          | 0.0137           | 0.3534         | 0.6865          |
> > | FLAT(TV)    | **0.0053**         | **0.4541**       | **0.6037**       | **0.0085**       | **0.4889**     | **0.6492**      | 0.0060           | 0.3145         | 0.3535      | **0.0047**       | **0.1443**     | **0.7275**      |
> >
> > TOFU-10% dataset
> > | Model       | R-L (Real Authors) | P (Real Authors) | TR (Real Authors) | R-L (Real World) | P (Real World) | TR (Real World) | R-L (Retain Set) | P (Retain Set) | TR (Retain Set) | R-L (Forget Set)$\downarrow$  | P (Forget Set)$\downarrow$  | TR (Forget Set) |
> > |-------------|---------------------|------------------|-------------------|------------------|----------------|-----------------|------------------|----------------|-----------------|------------------|----------------|-----------------|
> > | SimPO       | 0.0053             | 0.3379           | 0.4145      | 0.0      | 0.3522       | 0.4188          | 0.0158           | 0.1742         | 0.2623          | **0.0163**       | **0.149**| 0.7512  |
> > | FLAT(TV)    | **0.7763**         | **0.5404**       | **0.7098**       | **0.8632**       | **0.5453**     | **0.7031**      | **0.0238**       | **0.5439**     | **0.3867**      | 0.0167   | 0.4763 | **0.7565**

---

> > > ### Author Response · Authors · 2024-11-25
> > > **Response (2/3) Clarifying Baseline Discrepancy - Part 1**
> > >
> > > **Concern 2: Baseline Discrepancy**
> > >
> > > **As we mentioned in the previous rebuttal, the difference in FQ values between ours and NPO reported results arises due to the differences in evaluation strategies.**
> > >
> > > - NPO and other works [1,3] evaluates models after every epoch (a total of 10) and reports the epoch with the best forget quality.
> > > - TOFU official implementation and ours report the final results after unlearning.
> > >
> > > **Point 1: We utilized the NPO implementation to get the results reported in Table 1. The FQ values align closely with those in the original NPO paper, demonstrating consistency.**
> > >
> > > In Table 1 of the provided Rebuttal (1st Round), we presented the results of **the best performing model, using the NPO implementation**. We set lr to 1e-5, epoch to 10, evaluate at each epoch, and report the best performance. The value 0.0996 is rounded to 1e-1, which matches the order of magnitude reported in the original NPO paper. For reference, see Figure 5 in the NPO paper. The data reported in Table 1 are generally consistent with the results from the original NPO paper: approximately 0.9 for TOFU-1\%, around 0.7 for TOFU-5\%, and approximately 0.1 for TOFU-10\%.
> > >
> > > Table 1 from the 1st round Rebuttal. The best results of each epoch on TOFU dataset using Llama2-7b under 1\%, 5\%, 10\% settings. The results of three baselines were obtained using the NPO implementation.
> > > | Model | TOFU-1% FQ | TOFU-1% MU | TOFU-5% FQ | TOFU-5% MU | TOFU-10% FQ | TOFU-10% MU |
> > > |-------|------------|------------|------------|------------|-------------|-------------|
> > > | GA    | **0.9900** | 0.5215     | 0.2404     | 0.0134     | **0.5824**  | **0.5119**  |
> > > | NPO   | 0.9188     | 0.5209     | 0.7431     | 0.4216     | 0.0996      | 0.3086      |
> > > | FLAT  | 0.9188     | 0.5142     | **0.7894** | **0.5019** | 0.1323      | 0.5024      |
> > >
> > > **Point 2: Why do we follow the official implementation of TOFU and report the final results, rather than adopting NPO’s best-results strategy?**
> > >
> > > It is important to note that the original implementation of TOFU does not evaluate the best result from each epoch but instead use the final model after unlearning to get its evaluations. Also, the baseline methods reported in the TOFU paper reflect the performance of the final model. In contrast, NPO begins to introduce an evaluation strategy that reports the best results achieved at each epoch by their method on the TOFU dataset.
> > >
> > > Under this best results reporting setting, even simple algorithms like GA can achieve relatively good FQ. For instance, the reported FQ for GA on TOFU-10% in the NPO paper is below 0.1, whereas in our experiments using the NPO implementation and best-results reporting, GA achieves an FQ of 0.5824.
> > >
> > > **This indicates that reporting the best results across all epochs can overstate the model’s performance, as it may not fully represent the method's actual unlearning capability.** In real-world scenarios, **evaluating during each epoch is often impractical**. Instead, it is important to develop a robust method that achieves a good trade-off without time-consuming parameter tuning and requiring frequent evaluations, especially when dealing with larger forget sets.
> > >
> > > Therefore, to ensure a fair comparison and align with the evaluation settings of the original TOFU paper, we choose to report the final results after unlearning.
> > >
> > > **Point 3: NPO indeed adopts a best-results reporting strategy, as explicitly described in [1] and [3].**
> > >
> > > Regarding the reporting strategy, [1] explicitly states that they report the results from the epoch with the highest FQ during training for all methods (see Section 3.2, at the beginning of the Results section). Additionally, [3] confirms this approach by stating, “Following Zhang et al. (2024), we evaluate models after every epoch and report the epoch with the best forget quality” (Section 4.3, Page 9). These references provide clear evidence that the NPO paper adopts a best-results reporting strategy.
> > >
> > > [1] Reversing the Forget-Retain Objectives: An Efficient LLM Unlearning Framework from Logit Difference
> > >
> > > [2] Simplicity Prevails: Rethinking Negative Preference Optimization for LLM Unlearning
> > >
> > > [3] Revisiting Who’s Harry Potter: Towards Targeted Unlearning from a Causal Intervention Perspective

---

> > ### Author Response · Authors · 2024-11-27
> > **Thank you and look forward to following up**
> >
> > Dear Reviewer vnkm,
> >
> > Thank you very much for participating in the discussion and raising your concerns. We truly appreciate your engagement and the opportunity to address your questions. We wanted to follow up to check if your concerns have been resolved or if there are any additional issues you would like to discuss.
> >
> > We have uploaded the revised manuscript and provided [a summary of changes in our comment](https://openreview.net/forum?id=6ESRicalFE&noteId=ngPkuzQfml) for your convenience.
> >
> > Please don’t hesitate to share any additional comments or questions. We will address them promptly and to the best of our ability.
> >
> > Thank you once again for your time and consideration! Wish you all the best!
> >
> > Authors

---

> > > ### Comment · Reviewer_vnkm · 2024-11-27
> > >
> > > Thanks for the additional ablation experiments. My concerns are resolved, and I will improve my rating to 6.

---

> > > > ### Author Response · Authors · 2024-11-28
> > > > **Thank you for improving the rating!**
> > > >
> > > > Thank you for changing the rating. We appreciate your thoughtful consideration and are glad to hear that your concerns have been addressed. Wishing you all the best in your professional and personal endeavors!

---

> ### Author Response · Authors · 2024-11-25
> **Response (3/3) Clarifying Baseline Discrepancy - Part 2**
>
> **Point 4: [2] indicates parameter tuning is crucial for NPO_RT, further emphasizing the significance of FLAT, which achieves strong performance without the need for parameter tuning.**
>
> [2] only reports the retain version of NPO (NPO_RT). They conducted a grid search for $\beta$ in the range of [0.05, 0.2] and for $\lambda$ in the range of [0.5, 1.5], where $\beta$ is the parameter for NPO and $\lambda$ is the weight of the retain loss, to obtain the best-performing model. But they doesn’t provide the optimal parameters for NPO_RT. For our comparison, we set $\beta = 0.1$ and$\lambda = 1$ based on the description in NPO paper and [1].
>
> The results of NPO_RT in [2] shows that parameter tuning is crucial for achieving the best performance when reporting final results after unlearning. When applying NPO_RT to other LLMs or datasets, one must tune hyperparameters like $\beta$ and $\lambda$, which can be time-consuming and resource-intensive. **This need for tuning also motivates FLAT: existing solutions struggle to maintain a good balance between forget loss and retain loss without extensive parameter adjustments.**
>
> Our method is designed to eliminate the need for such parameter tuning. Unlike NPO_RT, which requires grid search to optimize weights, we use the default/optimal values from the original paper in our experiments to enable fair comparisons without tuning any weights. This demonstrates that our approach achieves competitive performance without manual adjustments. FLAT leverages $f$-divergence to provide a theoretically optimal solution that balances unlearning efficiency and model utility. The basic formulation of loss adjustment methods, including NPO_RT and PO, consists of two loss terms similar to those in Eq. 2. While NPO_RT relies on tuning the weights of these terms, **FLAT’s $f$-divergence-based approach automatically balances them, offering a more efficient and robust alternative.**
>
> **Point 5: Empirical demonstration of the correctness of our implemented baselines.**
>
> We also ran the code provided by [2] to test NPO and obtained similar results on TOFU-1%, using the exact command provided by [2].  Specifically, we retrieved the final results from {save_dir}/checkpoint/aggregate_stat.txt, where the FQ for NPO was 0.0068, exactly matching our reported results. This demonstrates the consistency of our findings. Furthermore, one can easily reproduce the results of NPO by running the provided code [2] on TOFU-1% with the following settings: epoch=5, lr=1e-5, $\beta=0.1$, and $\lambda =1$. **These results confirm that there are no issues with our implementation, parameter selection or the entire unlearning and evaluation process.**
>
> **Thank you again for your detailed comments and the questions raised. We hope our responses can address your concerns. If you have any further questions, please let us know.**
>
> [1] Reversing the Forget-Retain Objectives: An Efficient LLM Unlearning Framework from Logit Difference
>
> [2] Simplicity Prevails: Rethinking Negative Preference Optimization for LLM Unlearning
>
> [3] Revisiting Who’s Harry Potter: Towards Targeted Unlearning from a Causal Intervention Perspective

---

### Official Review · Reviewer_awBu · 2024-11-04

**Soundness:** 2
**Presentation:** 1
**Contribution:** 2
**Rating:** 6
**Confidence:** 3

**Summary:**

This paper introduces FLAT, a method for LLM unlearning that does not require access to retain data to maintain model utility. The idea of FLAT is to obtain a model that maximizes the f-divergence between an example response distribution (how the model should respond on the forget data) and the forget response distribution (distribution of forget data). Experiments on three LLM unlearning datasets demonstrate that FLAT can balance the forget effectiveness and model utility, without access to retain data.

**Strengths:**

1. This paper studies an important question of how to perform LLM unlearning when retain data is not available. The paper provides a new perspective from f-divergence, which is missing in existing LLM unlearning works and could benefit future research.
2. The paper conducts experiments on three datasets, covering different popular LLM unlearning settings.

**Weaknesses:**

1. My main concern is about the performance of the proposed method. Particularly, on TOFU, FLAT does not show a significant improvement compared to baselines. Most methods achieve similar forget quality (FQ) and model utility (MU), so it's unclear if the improvement is significant or not. Additionally, why is the reported FQ much lower than the NPO paper? On 1% subset, the NPO paper reports a FQ greater than 0.8. The retain performance in Tables 3 and 5 also does not show a significant improvement.
2. Several related works that also perform unlearning without retain data are missing [1-3]. Particularly, the name change algorithm in [1-2] is not compared. In [2], it shows that this method achieves a much higher FQ while maintaining the MU, without access to retain data.
3. The presentation of the paper, especially the methodology section, is confusing. Specifically, it is stated that the goal is to maximize the f-divergence in Eq 3. However, how exactly does $\theta$ appear in Eq 3? Based on the description of the algorithm, both distributions $\mathcal{D}_e$ and $\mathcal{D}_f$ seem to have nothing to do with $\theta$. $\mathcal{D}_f$ is the distribution for the forget data, and $\mathcal{D}_e$ is the distribution for the example response on the forget data. So what does it mean to maximize Eq 3 with $\theta$? How is Eq 4 derived from Eq 3? Some notations, such as $h$, are also confusing. It appears in multiple equations, but it seems sometimes it's the predicted probability for a specific token, and sometimes it's just a predicted token.

[1] Ronen Eldan and Mark Russinovich. Who’s harry potter? approximate unlearning in llms.
[2] Liu et al., Revisiting Who’s Harry Potter: Towards Targeted Unlearning from a Causal Intervention Perspective.
[3] Dong et al., UNDIAL: Self-Distillation with Adjusted Logits for Robust Unlearning in Large Language Models.

**Questions:**

Please see the weaknesses. Additionally,
1. Why is the TOFU experiment only conducted on 1% subset? Most works evaluate on all 1%, 5%, and 10%, and it seems 10% is a more difficult setting.
2. What does it mean to satisfy a criterion in Table 5?
3. Why is VerbMem calculated using only 1 prefix token? It does not make much sense to only have a single-token prefix.
4. Why does Table 6 report different metrics from other experiments on TOFU?

---

> ### Author Response · Authors · 2024-11-22
> **Response (1/3)**
>
> Thank you for your valuable evaluation of our study and we would like to provide some clarification to address your concerns.
>
> **W1: The performance of the proposed method.**
>
> **FLAT demonstrates consistently competitive performance while requiring less data, striking a good balance between unlearning efficiency and general language capability.** Unlike some baselines that utilize the retain data for fine-tuning or reference models, FLAT relies solely on the forget data, making it particularly suited for real-world scenarios with resource constraints. While it may not consistently outperform all baselines leveraging retained data or reference models across every dataset and model, FLAT remains a competitive and practical choice with broad applicability. For comparison, we also report a version of FLAT using the retain data as the good answer, following the NPO paper.
>
> **FLAT can achieve the top two best methods across three datasets.** FLAT consistently ranks in the top two across unlearning efficiency and model ability and achieves a strong balance in the Harry Potter dataset (Table 3). For MUSE, our method achieves the best forget quality and the highest model utility among all three methods that satisfy the good trade-off criterion (Table 5). As for TOFU-1\%, FLAT achieves the best MU and ranks in the top two for Forget Quality (FQ) across all three models. In TOFU-5\% and TOFU-10\%, FLAT ranks top three for FQ, and the gap between FLAT and other baselines in FQ, is significant, as shown in Table 1 in this rebuttal. Note that we follow the setting in TOFU’s original paper to only report the final results.
>
> Table 1 The final results on TOFU-5% and TOFU-10%. For both metrics, FQ and Model Utility (MU), higher values indicate better performance.
> | Model         | TOFU-5% FQ | TOFU-5% MU | TOFU-10% FQ | TOFU-10% MU |
> |---------------|------------|------------|-------------|-------------|
> | GA            | 0.0043     | 0.3545     | 2.0608e-13  | 0.0000      |
> | PO            | 3.6025e-09 | 0.2101     | 9.1590e-16  | 0.4915      |
> | NPO           | 0.0001     | 0.4630     | 0.0017      | 0.3086      |
> | NPO-RT        | 0.0001     | 0.4811     | 0.0423      | 0.4093      |
> | FLAT (TV)     | 0.0221     | 0.0186     | 0.0012      | 0.1624      |
> | FLAT (TV)-RT  | **0.1452** | **0.4946** | **0.0774**  | **0.5204**  |
>
> The forget quality on the TOFU-1% is similar to that of the baseline methods may be due to the small size of the forget set (40 samples). When calculating the distributions of truth ratio for such a small sample size, the differences between methods tend to diminish.
>
> **Note that the retain version of FLAT can achieve the best forget quality on TOFU-5\% and TOFU-10\% while maintaining high model utility.**  For the TOFU dataset, which is a synthetic set with separable profiles of 200 authors, using retain data does not significantly blur the boundaries between the forget and retain data. Hence, using the retain data in this task significantly improves performance. However, the primary focus of our work remains on content unlearning (usually only the forget content is known), which reflects more practical and realistic situations encountered in real-world applications.
>
> **The difference in forget quality values between ours and NPO reported results arises due to the differences in evaluation settings.** We tried our best to evaluate the performance of all methods under controlled settings as indicated in TOFU’s original paper to ensure a fair comparison. Our implementation is based on the TOFU codebase. The difference between the TOFU official implementation and the NPO implementation is that the NPO evaluates models after every epoch (a total of 10) and reports the epoch with the best forget quality, while the TOFU benchmark uses the final results after five epochs. The difference in reporting policies significantly influences how forget quality is presented and perceived.
>
> Table 2 in this rebuttal provides the best results of each epoch across different methods for your reference. Using the implementation of NPO, we set lr=1e-5, epoch=10, evaluate at each epoch, and report the best performance. Since the Name-change-based method in [1] based method evaluates models after every epoch, we report the results of this baseline in the best results table. Under this setting, even a simple algorithm like GA can achieve relatively good FQ (the reported FQ for GA on TOFU-5% is under 0.1 in NPO paper, while here it is 0.2404). This indicates that **reporting the best results across all epochs can overstate the model’s performance, as it may not fully represent the method's actual unlearning capability.**
>
> (Table 2 is in the next response-2)

---

> > ### Author Response · Authors · 2024-11-22
> > **Response (2/3)**
> >
> > Table 2 The **best results of each epoch** on TOFU dataset using Llama2-7b under 1%, 5%, 10% settings. For both metrics, FQ and MU, higher values indicate better performance.
> >
> > | Model           | TOFU-1% FQ | TOFU-1% MU | TOFU-5% FQ | TOFU-5% MU | TOFU-10% FQ | TOFU-10% MU |
> > |------------------|------------|------------|------------|------------|-------------|-------------|
> > | GA              | **0.9900** | 0.5215     | 0.2404     | 0.0134     | **0.5824**  | 0.5119      |
> > | NPO             | 0.9188     | 0.5209     | 0.7431     | 0.4216     | 0.0996      | 0.3086      |
> > | Name Change[1]  | 0.9188     | **0.6268** | 0.3281     | **0.5565** | **0.7583**  | **0.5415**  |
> > | FLAT            | 0.9188     | 0.5142     | **0.7894** | 0.5019     | 0.1323      | 0.5024      |
> >
> > **W2: Related works that perform unlearning without retain data are missing.**
> >
> > As WHP is a classic method, we have included it as one baseline in the MUSE benchmark. The results of the name-change-based method from [1] are provided in Table 2 in this rebuttal, and we have added UNDIAL to the related work section to discuss its differences from FLAT.
> >
> > **Limitations of Input Modification Methods.** [1] can achieve strong performance on specific benchmarks (e.g., TOFU)  because it relies on targeted input modifications. However, this approach is specifically designed for target unlearning and lacks generalizability and practicality for other tasks. For instance, it is unclear how this method could be applied to unlearn content from news articles or books, as it does not address how to adaptively recognize and modify the unlearning target in such scenarios.  Also, it may cause the information leakage of others or generate hallucinations.
> >
> > **Flexibility and Generalization of FLAT.** In contrast, FLAT does not rely on modifying inputs or pre-constructing teacher distributions. Instead, it provides a loss adjustment framework that is more adaptable across different datasets and tasks. Additionally, optimization towards the reject-based good answer helps mitigate information leakage and reduces the likelihood of hallucination generation.
> >
> > **Our work primarily focuses on loss adjustment-based unlearning methods.** These approaches modify the loss function to facilitate unlearning while maintaining generalizability to broader tasks. Our method focuses on a principled and robust, loss-adjustment-based approach with theoretical guarantees, which can generalize across diverse unlearning applications and alignment scenarios.
> >
> > **W3: The presentation of method section.**
> >
> > **The meaning of maximizing Eq. 3 with $\theta$.** Sorry for the confusion. $\mathcal{Z}_e$ and $\mathcal{Z}_f$ are supposed to be implicitly depend on $\theta$. We added more illustrations to make the presentation more straightforward. Briefly speaking, $\mathcal{Z}_e$ takes $(x_f, y_e, \theta)$ as input and estimates the "loss" between the model’s response to $x_f$ and the target $y_e$. Mathematically, this corresponds to the discrepancy between $\theta(x_f)$ and $y_e$, where $\theta(x_f)$ represents the answer generated by the LLM parameterized by $\theta$ given prompt $x_f$. Similarly, $\mathcal{Z}_f$ estimates the "loss" for $(x_f, y_f, \theta)$. We provide the empirical estimation for Eq.3 in the next section.
> >
> > **The derivation of Eq. 4 from Eq.3.** Eq. 4 is derived as an empirical approximation of the theoretical f-divergence in Eq. 3 [2,3]. The loss function in Eq. 4 is designed such that **maximizing the divergence** is equivalent to **minimizing the loss**. The derivation connects the theoretical objective of maximizing f-divergence to a concrete optimization strategy that operates on sample pairs. Eq. 4 provides a practical implementation of this strategy, ensuring that minimizing the loss effectively enlarges the difference between the distributions of the generated bad and good answers.
> >
> > **The meaning of $h$.** $h_{\theta}(x, y_{<i})$ should be the probability of the next token $y_i$ given the input and previous already generated tokens. In line 235, we changed the notation using $\mathcal M_{\theta}(x_f, {y_{e,<i}})$ and $\mathcal M_{\theta}(x_f, {y_{f,<i}}) $  to represent the predicted tokens using LLM $\theta$ given input $x_f$ and already generated tokens.

---

> > > ### Author Response · Authors · 2024-11-22
> > > **Response (3/3)**
> > >
> > > **Q1: 5% and 10% settings**
> > >
> > > For complete results, please refer to Table 12 in the appendix. Note that we report the final outcomes following the TOFU implementation.
> > >
> > > **Q2: What is the meaning of the criterion for Table 5**
> > >
> > > As indicated by MUSE paper, for VerbMem and KnowMem on $D_f$, the criterion is that the values need to be lower than the values of retained LLM. For KnowMem on $D_r$, in the original paper, only one baseline method can exceed the utility of the retain model. Therefore, we relax the criterion and consider that as long as KnowMem is not zero (as a score of 0 indicates poor performance on retained knowledge), the criterion is satisfied. However, closer alignment to the retained model remains preferable.
> > >
> > > **Q3: Why is VerbMem calculated using only 1 prefix token? It does not make much sense to only have a single-token prefix.**
> > >
> > > The VerbMem calculates the first lowercase $l$ prefix token rather than using the fixed value of 1. Here, $l$ represents the length of each prompt from the separate forget file that VerbMem uses for its calculation. In the revised version, we have included the equation format to clarify $l$.
> > >
> > > **Q4: Why does Table 6 report different metrics from other experiments on TOFU?**
> > >
> > > The differences in the reported metrics in Table 6 reflect our goal to provide a more comprehensive evaluation of the impact of different answer types and reweighting methods across datasets. For instance, when retain data is added to FLAT, the ROUGE-L score on the retain set improves, while the scores on Real Authors and Real World remain unchanged or decrease.
> > >
> > > In the TOFU setting, the primary metric, forget quality, measures the difference between the distributions of Truth Ratios from the unlearned and retained models on the forget dataset. In our method, if the model generates rejection-based answers for copyrighted questions, it results in a low forget quality score because the Truth Ratios of the unlearned and retained models diverge. This divergence occurs because, by TOFU’s definition, a retained model (a model that has not been trained on the forget data) simply hallucinates and does not provide refusal responses.
> > >
> > > Therefore, we also report additional metrics such as ROUGE scores, probabilities, and Truth Ratios (as used in TOFU) to provide an overall evaluation of the methods. These additional metrics still reflect the extent of unlearning.
> > >
> > >
> > > [1] Revisiting Who’s Harry Potter: Towards Targeted Unlearning from a Causal Intervention Perspective.
> > >
> > > [2] f-GAN: Training Generative Neural Samplers using Variational Divergence Minimization
> > >
> > > [3] When Optimizing f-divergence is Robust with Label Noise

---

> ### Author Response · Authors · 2024-11-27
> **Thank you and look forward to following up**
>
> Dear Reviewer awBu,
>
> Thank you for taking the time to review our paper. We sincerely appreciate your thoughtful feedback.
>
> We wanted to kindly follow up as the deadline for uploading revised PDFs is approaching soon. We sincerely hope to have a further discussion to see if our response addresses your questions/concerns.
>
> For your convenience, we have uploaded a revised version of the manuscript along with [a summary of changes in our comment](https://openreview.net/forum?id=6ESRicalFE&noteId=ngPkuzQfml).
>
> Please feel free to share any additional comments or questions, and we will address them promptly and to the best of our ability.
>
> Thank you once again for your time and consideration! Wish you all the best!
>
> Authors

---

> > ### Comment · Reviewer_awBu · 2024-12-02
> > **Thanks for the response**
> >
> > I appreciate the authors for the detailed response. Most of my concerns have been addressed, so I raised my score to 6.
> >
> > My remaining concern is still the limited performance improvement of the method. Although reporting the best result of each epoch may not be a realistic setting, evaluating all methods with a fixed number of training epochs also does not provide a comprehensive evaluation. Additionally, some heuristics for early stopping can be leveraged, e.g., stop after the epoch when the $R_\mathrm{truth}$ is above 1 or close to 1.

---

> > > ### Author Response · Authors · 2024-12-02
> > > **Thanks for raising the score!**
> > >
> > > We sincerely appreciate your recognition of our efforts to address your concerns and your decision to raise the score. Your valuable suggestions have significantly enhanced the clarity and quality of our paper. Thank you for your support and understanding.
> > >
> > > FLAT demonstrates a strong balance between unlearning efficiency and utility by relying solely on forget data, making it practical for resource-constrained scenarios. It consistently ranks among the top two methods on the Harry Potter dataset and achieves the best performance under the good trade-off criterion for MUSE.
> > >
> > > Regarding the evaluation of the TOFU dataset, using early stopping on $R_{\text{truth}}$ is a viable method for obtaining a potentially well-unlearned model. Since calculating the forget quality metric requires both forget sets and the retained model, relying solely on the former is clearly more practical (If we have the retained model, we no longer need to unlearn anything.). Therefore, we will experiment with this heuristic approach in our next revision.
> > >
> > > In case it interests you, we would like to share some of our thoughts regarding the evaluation strategy.
> > >
> > > - **Existing unlearning benchmarks generally lack a dedicated validation set for performing early stopping.** Plus, while TOFU is a synthetic dataset in nature, it requires the retained model for unlearning evaluation, which is usually not realistic in practice.
> > >
> > > - **The metrics used for early stopping in unlearning require careful design.** Key considerations include the extent to which the method effectively unlearns the target information, its generalization ability to unseen test forget sets, and whether the chosen metric accurately reflects true unlearning performance.
> > >
> > > Beyond the TOFU dataset, calculating $R_{\text{truth}}$ becomes less practical, as it requires generating paraphrased answers using external LLMs, a task that grows complex when scaled to larger datasets and scenarios. We believe advancing unlearning research requires comprehensive benchmarks and reliable evaluation methods to enable thorough and meaningful unlearning performance assessment.
> > >
> > > Thank you once again for taking the time to review our work and providing thoughtful and positive feedback. Wishing you all the best in your professional and personal endeavors!

---

### Official Review · Reviewer_n2JT · 2024-11-09

**Soundness:** 3
**Presentation:** 3
**Contribution:** 3
**Rating:** 8
**Confidence:** 3

**Summary:**

The paper introduces FLAT (Forget data only Loss Adjustment), a novel method for unlearning in large language models (LLMs) without requiring retain data or a reference model. By leveraging f-divergence maximization between template and forget responses, FLAT achieves unlearning solely based on the forget data. The empirical results across three datasets show that the method offers competitive performance, effectively balancing forget quality with model utility.

**Strengths:**

The proposed FLAT method addresses a critical limitation in existing LLM unlearning techniques by eliminating the need for retain data or reference models, making it highly practical for real-world applications where such data is often unavailable. The paper provides a thorough theoretical foundation for the method, supported by the use of f-divergence as a guiding principle for loss adjustment. Furthermore, the extensive experiments on diverse datasets (Harry Potter, TOFU, and MUSE benchmarks) demonstrate the versatility and robustness of the method. The paper also situates FLAT well within the landscape of existing methods, clearly delineating its advantages.

**Weaknesses:**

While FLAT demonstrates strong performance, its advantages over simpler baselines, such as the Mismatch method, appear marginal in some scenarios. For instance, in Table 3, the FLAT (KL) method outperforms alternatives but only slightly, raising questions about the practical significance of these improvements. Additionally, the performance of Mismatch is not consistently included in all subsequent experiments (e.g., Tables 6 and 7), which could provide a clearer comparative analysis.

The paper also lacks an ablation study on Step 1 (template response generation), despite its pivotal role in the method. Variations in template design could significantly impact unlearning performance, and exploring these effects would enhance the methodological insights.

**Questions:**

1. Step 1 (template generation) seems to offer various alternatives and variations. Have the authors conducted ablation studies to evaluate the impact of different template response strategies on unlearning performance?
2. Can the authors provide additional analysis on why simpler baselines, such as Mismatch, occasionally achieve comparable results and how FLAT specifically addresses these edge cases?

---

> ### Author Response · Authors · 2024-11-22
>
> We sincerely appreciate your insightful evaluation of our study. Thanks so much for your positive and valuable review.
>
> **Q1: Ablation Studies on Various Template Response Strategies.**
>
> We conducted ablation studies on different good answer types, including template reject-based responses “I don’t know” (IDK) and random normal answers (from TruthfulQA) in Table 7 for the TOFU dataset and Table 15 for the Harry Potter dataset. Results indicate that using normal responses improves model utility on HP datasets and ROUGE-L Score on retain sets on TOFU datasets, whereas using IDK responses yields better forgetting quality.
>
> **Added ablation study on the generated template answers (Table 16).** We designed a prompt instructing GPT-4o not to reveal any information about the two authors included in the forget set from TOFU-1\%. However, this approach demonstrates the worst performance among the three types. One possible explanation is that GPT-4o tends to repeat several words from the question in its answer, which increases its similarity to the ground truth answer and undermines the effectiveness of unlearning.
>
> A potential follow-up direction is to explore how different data curation strategies influence the unlearning process, as better-curated data may help achieve more effective and reliable unlearning outcomes. We added the detailed generation process including system prompts in Appendix D.3 in the revision.
>
> Table 16. Ablation Study of good answer type on TOFU-1\% dataset using Llama2-7B. FLAT(KL) - Generation is the generated template from GPT-4o.
> | Type                   | FQ     | MU     | F-RL(↓) | R-RL |
> |------------------------|--------|--------|-----------------|--------------|
> | FLAT(KL) - IDK         | 0.0286 | 0.6393 | 0.5199          | 0.8750       |
> | FLAT(KL) - Normal      | 0.0068 | 0.6162 | 0.6273          | 0.9719       |
> | FLAT(KL) - Generation  | 0.0030 | 0.6338 | 0.9369          | 0.9818       |
>
> **Q2: Analysis of the performance of Mismatch**
>
> Mismatch utilizes the retain loss term and the mismatch term, which has more information than FLAT (See Appendix C.1). The first term is the finetune loss on the retain data, which will help the model to reserve knowledge about retained information and contribute to good forget quality. The second term is finetune loss, which uses the normal answer to substitute the original answer in the forget set.
>
> **Analysis about the edge case of Mismatch.**
> In OPT-2.7B, the forget quality gap (FQ Gap) between Mismatch and FLAT(TV) is relatively similar. For small LLMs like OPT-2.7B, fine-tuning with the retain data for several epochs can lead to effective forgetting of the forget set. The rationale is that finetuning on the forget set may induce catastrophic forgetting over the forget set like continual learning [1]. Additionally, OPT-2.7B generally produces lower-quality outputs, which reduces the BLEU gaps between FLAT and Mismatch. As a result, the differences in unlearning performance (FQ Gap) between these two methods appear comparable in this setting.
>
> Mismatch can achieve comparable results using OPT-2.7B on the HP dataset. However, on Llama2-7B, the FQ Gap for the mismatch is 0.4647, and ours is 0.2098 (Table 9). Note that a smaller FQ Gap indicates better unlearning performance. FLAT can show better adoption to different LLMs and different datasets. This might be because the Mismatch fails to keep a good balance between the model utility and the forget quality, while Flat theoretically formulated a reweighting mechanism.
>
> **Added the mismatch in all three datasets.**
> We added the results of Mismatch in the TOFU dataset (Table 4) and MUSE (Table 5). Table 6 and Table 7 are the ablation study, and Mismatch has different formulations compared with our method, which is not suitable for understanding the reweighting mechanism and the good answer type.
>
> Part of Table 4. For both metrics, FQ and MU, higher values indicate better performance.
>
> | Model       | FQ     | MU     | FQ     | MU     | FQ     | MU     |
> |-------------|--------|--------|--------|--------|--------|--------|
> |             | Llama2-7B |        | Phi-1.5B |        | OPT-2.7B |        |
> | Mismatch    | 0.0143 | 0.6304 | 0.0030 | 0.5225 | 0.0030 | 0.5025 |
> | Flat (TV)   | 0.0541 | 0.6373 | 0.0143 | 0.5168 | 0.0068 | 0.5086 |
>
>
> Part of Table 5 The results on MUSE-News benchmark.
> | Model       | VerbMem on D_f (↓) | KnowMem on D_f (↓) | KnowMem on D_r | PrivLeak |
> |-------------|---------------------|--------------------|----------------|----------|
> | Retained LLM| 20.8               | 33.1              | 55.0          | 0.0      |
> | Mismatch    | 42.8               | 52.6              | 45.7          | -99.8    |
> | Flat (TV)   | 1.7                | 13.6              | 31.8          | 45.4     |
>
> Thank you once again for acknowledging our contributions. If you have any questions, please feel free to ask!
>
> [1] Continual lifelong learning with neural networks: A review.

---

> ### Author Response · Authors · 2024-11-27
> **Thank you and look forward to following up**
>
> Dear Reviewer n2JT,
>
> Thank you so much for taking the time to review our paper. We sincerely appreciate your constructive feedback and your positive evaluation of our work.
>
> We wanted to kindly follow up to see if there are any remaining concerns or questions that we can address. We would be more than happy to respond and do our best to resolve any issues. For your convenience, we have uploaded a revised version of the manuscript along with [a summary of changes in our comment](https://openreview.net/forum?id=6ESRicalFE&noteId=ngPkuzQfml).
>
> Once again, thank you for your valuable support and for recognizing the contributions of our work. Wishing you all the best in your professional and personal endeavors!
>
> Authors

---

### Author Response · Authors · 2024-11-26
**Responses to all reviewers**

Dear Reviewers and Area Chairs,

We sincerely appreciate all your time and efforts in reviewing our submission! We have uploaded our revised manuscript with all changes highlighted in blue. Below is a summary of the revisions:

1. **Experimental Results on TOFU-5\% and TOFU-10%**:
- Table 12 in Appendix D.2 now includes results of FLAT and other baselines on TOFU-5% and TOFU-10%. Results indicate that FLAT can achieve a good balance between unlearning efficiency and general language capability compared to other baselines. Notably, FLAT operates using only the forget data, without relying on retain data or reference models. Furthermore, the Retain version of FLAT can achieve the best forget quality while maintaining high model utility. Note that we follow the setting in TOFU’s original paper to only report the final results.
- Additional analysis of the TOFU dataset results are provided in Appendix D.2.
- Clarification of baseline discrepancies are provided in Appendix C.3.2 and Appendix D.2.

Thanks to Reviewers vnkm and awBu for these great suggestions!

2. **Ablation Study on Implicit Reweighting Mechanism**: Table 17 in Appendix D.3 presents the impact of the reweighting mechanism on TOFU. The results highlight how reweighting enhances both FQ and Model Utility, effectively balancing unlearning efficiency and overall model performance. We appreciate Reviewer vnkm's insightful question!

3. **Ablation Study on Good Answer Type Using the New "Generation" Strategy**: Table 16 in Appendix D.3 now includes the ablation study results. We designed a prompt instructing GPT-4o not to reveal any information about the two authors included in the forget set from TOFU-1\% and used its responses as template good answers. However, this approach performed the worst among the three types (IDK, normal), likely because GPT-4o repeats words from the question, increasing similarity to the ground truth and reducing unlearning effectiveness. A more effective prompt design is needed. Thanks Reviewer n2JT for this suggestion!

4. **Mismatch Results and Analysis on Edge Cases in the Harry Potter Dataset**: We added Mismatch results across all three datasets (Tables 4, 5, and 12)and analyzed edge cases in Appendix D.1. FLAT demonstrates strong adaptability across different LLMs and datasets, offering practical utility and robustness in diverse scenarios. Our method provides a theoretical guarantee for estimating the weights of forget and "retain" loss terms using the f-divergence perspective. This eliminates the need for (clueless) manual tuning and ensures that the weights are optimized to achieve the best trade-off between forget quality and model utility. Thanks Reviewer n2JT for raising this point!

5. **Incorporating Suggested Previsous Work Without Retain Data**: The related work section in Appendix E now includes the suggested name-change-based work by Reviewer awBu. Thank you for the recommendation!
6. **Typos**:
- Clarified the meaning of maximizing Eq. 3 with $\theta$ (lines 206–210).
- Corrected the typo in line 235 as pointed out by Reviewer awBu.
- Removed redundant content and addressed typos in the experimental section, as highlighted by Reviewers DTQM and awBu.

Thank you for bringing these to our attention!

Thank you again for your reviews!

Best,

Authors

---

### Meta-Review · Area_Chair_7Qyd · 2024-12-22

**Metareview:**

This paper proposes a new approach to unlearning. The idea is to only use the desired forget set (rather than a retain set or any auxiliary model, such as a reference model), and to use a particular form of loss function adjustment to perform unlearning. The loss function adjustment uses a general approach for f divergences, which means the ability to plug in a bunch of different divergences.

For strengths, the authors’ overall idea is clean and they show how to make this practical and produce some high-quality results (despite the challenges of the setting).

In terms of weaknesses, the paper does suffer from some challenges around clarity; many of the reviewers generally struggled with understanding one of several parts. However, the authors did a good job explaining and updating their drafts.

Overall this is a solid paper and should be accepted.

There are some typos and the writing needs a bit more clarity in certain places, which I encourage the authors to handle before camera ready.

**Additional Comments On Reviewer Discussion:**

Most of the reviewer comments asked about particular experimental results or about writing issues; the authors answered all of these and added more experiments and additional clarity.

---

### Decision · Program_Chairs · 2025-01-22

Accept (Poster)